



# The imprints of contemporary mass redistribution on regional sea level and vertical land motion observations

Thomas Frederikse[1], Felix W. Landerer[1], and Lambert Caron[1]

[1]Jet Propulsion Laboratory, California Institute of Technology, Pasadena, California, USA

**Correspondence:** Thomas Frederikse (thomas.frederikse@jpl.nasa.gov)

**Abstract.** We derive trends and monthly anomalies in global and regional sea-level and solid-earth deformation that result from mass redistribution observed by GRACE and an ensemble of GIA models. With this ensemble, we do not only compute mean changes, but we also derive uncertainty estimates of all quantities.

We find that over the GRACE era, the trend in land mass change has led to a sea-level trend of 1.28-1.82 mm yr$^{-1}$, which
is driven by ice mass loss, while terrestrial water storage has increased over the GRACE period, causing a sea-level drop of 0.11-0.47 mm yr$^{-1}$. This redistribution of mass causes sea-level and deformation patterns that do not only vary in space, but also in time.

The temporal variations affect GNSS-derived vertical land motion (VLM) observations, which are now commonly used to correct tide-gauge observations. We find that for many GNSS stations, including GNSS stations in coastal locations, solid-earth
deformation resulting from present-day mass redistribution causes trends in the order of 1 mm yr$^{-1}$ or higher. Since GNSS records often only span a few years, these trends are generally not representative for the tide-gauge records, which often span multiple decades, and extrapolating them backwards in time could cause substantial biases.

To avoid this possible bias, we computed trends and associated uncertainties for 8228 GNSS stations after removing deformation due to GIA and present-day mass redistribution. With this separation, we are able to explain a large fraction of the
discrepancy between observed sea-level trends at multiple long tide-gauge records and the reconstructed global-mean sea-level trend from recent reconstructions.

*Copyright statement.* © 2018. California Institute of Technology. U.S. Government sponsorship acknowledged.

## 1   Introduction

Mass loss from glaciers and ice sheets (Bamber et al., 2018) and changes in terrestrial water storage (TWS, Reager et al.,
2016) have resulted in an increase of ocean mass and a rise of global sea level over the past decades (WCRP Global Sea Level Budget Group, 2018). This redistribution of mass over the earth surface causes substantial changes in the earth's gravity field, the rotation parameters, and it deforms the solid earth (Farrell, 1972; Clark and Lingle, 1977; Milne and Mitrovica, 1998). Due to these effects, mass redistribution results not only in global sea-level changes, but also in regional patterns of sea-level





change and solid-earth deformation ('deformation' from here on). The regional relative sea-level (RSL) patterns are observed by tide gauges (e.g. Galassi and Spada, 2017), while deformation is observed by permanent GNSS stations as vertical land motion (VLM, Pfeffer et al., 2017). A thorough understanding of the causes of these regional patterns of sea level and VLM is an important prerequisite to projecting future regional sea-level changes (Kopp et al., 2014) and to using local observations to
infer global sea-level changes (Dangendorf et al., 2017; Frederikse et al., 2018).

The Gravity Recovery and Climate Experiment (GRACE, Tapley, 2004) satellite mission has provided estimates of present-day mass redistribution (PDMR) over the earth surface with unprecedented resolution and accuracy (Rodell et al., 2018). The resulting regional RSL and deformation patterns can be directly computed from GRACE observations (Riva et al., 2010). In this paper, when we refer to 'RSL', we refer to RSL changes, not its mean value. Glacial isostatic adjustment (GIA) also
cause regional RSL and deformation patterns, and to obtain PDMR changes from GRACE, its observations must be corrected for GIA. However, the effects of GIA come with an uncertainty (e.g. Martín-Español et al., 2016), which in turn affects GRACE estimates and the resulting global and regional RSL and deformation patterns. We use GRACE observations and the large ensemble of GIA solutions from Caron et al. (2018) to derive PDMR estimates and the resulting global and regional deformation and RSL changes. We also derive robust uncertainties of these estimates.

We use these deformation estimates to derive VLM trends that are corrected for GIA and PDMR. VLM trends derived from GNSS time series are more and more often used to correct sea-level observations from tide gauges for the impact of local VLM (Bouin and Wöppelmann, 2010; King et al., 2012; Pfeffer and Allemand, 2016; Wöppelmann and Marcos, 2016). This correction has led to improved estimates of global and regional sea-level changes (Wöppelmann et al., 2014; Hamlington et al., 2016; Frederikse et al., 2016; Dangendorf et al., 2017). However, in most of these studies it is commonly assumed that the linear
VLM trend, derived over the short GNSS record, is representative for the full time span of the associated tide gauge, which often covers multiple decades. This assumption generally holds for glacial isostatic adjustment (GIA) outside low-viscosity regions (e.g. Whitehouse, 2018), sediment isostatic adjustment, and compaction (Ferrier et al., 2017), but for many processes, such as subsidence due to local groundwater depletion (Kolker et al., 2011; Minderhoud et al., 2017) and co-seismic and post-seismic activity (Segall and Davis, 1997), this assumption does not hold. As a result, correcting long tide-gauge records for the
VLM trend observed over the short GNSS record could introduce a bias (Featherstone et al., 2015).

PDMR could also be a source of such a bias. Both ice mass loss and TWS changes cause substantial deformation trends (Riva et al., 2017; Santamaría-Gómez and Mémin, 2015), and these trends explain a non-negligible part of VLM signals observed by permanent GNSS stations (Pfeffer et al., 2017; Schumacher et al., 2018). Due to the accelerating pace of ice-mass loss (Bamber et al., 2018) and the large decadal and multi-decadal TWS variability (Boening et al., 2012; Reager et al., 2016; Humphrey
et al., 2017), the resulting deformation cannot be described by a linear trend on long time scales, and trends over the GNSS time span are in general not representative for longer periods.

To avoid this possible bias, we compute VLM trends corrected for GIA and PDMR at a global network of 8,228 stations from the Nevada Geodetic Laboratory database (Blewitt et al., 2018). These VLM trends can be used to correct tide-gauge observations for local VLM while avoiding the aforementioned possible bias. Furthermore, these trends can be used for any
study that relies on VLM estimates, but for which deformation due to GIA and PDMR are unwanted signals. As an example





application, we apply the corrected VLM trends on a subset of the tide-gauge data set used in Thompson et al. (2016, T16). T16 find that many of the longest tide-gauge records show higher sea-level trends for the 20th century than estimated by the recent global reconstructions from Hay et al. (2015) and Dangendorf et al. (2017). We show that correcting these long-term tide-gauge records for local VLM explains a large part of this discrepancy.

## 2   Data and methods

In this section, we first introduce the estimates of GIA and PDMR in Paragraph 2.1 and 2.2. Then, we briefly discuss the methodology to compute the resulting regional RSL and deformation patterns in Paragraph 2.3. Finally, we discuss how these estimates are used to compute VLM trends from GNSS observations in Paragraph 2.4.

### 2.1   The GIA model

GIA affects GRACE observations, causes solid-earth deformation, which affects VLM observed by GNSS stations, and causes regional sea-level changes, which are observed by tide gauges (Tamisiea, 2011). To correct all these observations in a consistent way, and to derive uncertainties that result from the GIA correction, we use the GIA model ensemble from Caron et al. (2018). It does not only predict an estimate of the GIA-induced changes, but it also provides a large ensemble of predictions, computed from varying the solid-earth parameters and the ice-sheet histories. Each ensemble member comes with a likelihood that reflects its fitness to a global dataset of vertical GPS velocities and paleo sea-level records. From this ensemble, we can derive uncertainties for GIA-induced changes. We use a subset of 5,000 ensemble members from the original set that contains 128,000 GIA models. This number is sufficient to approach the original ensemble set with a maximum deviation of 2.5 percent from the original covariance matrix. Each ensemble member provides an estimate of the GIA signal observed by GRACE and the RSL and deformation patterns. PDMR estimates, and also the GIA correction that must be applied to GRACE observations to obtain PDMR, are often expressed as a change in equivalent water height (EWH).

From this ensemble, we compute mean trends $T_\mu$ and uncertainty $T_\sigma$, weighted by the likelihood of each solution, as follows:

$$T_\mu = \sum_{n=1}^{N} \frac{\mathcal{L}(n)}{\sum_{n=1}^{N} \mathcal{L}(n)} T(n)$$

$$T_\sigma = \sqrt{\sum_{n=1}^{N} \frac{\mathcal{L}(n)}{\sum_{n=1}^{N} \mathcal{L}(n)} \left(T(n) - T_\mu\right)^2},$$

where $N$ is the number of ensemble members, $T(n)$ the trend of the individual ensemble member, and $\mathcal{L}$ the likelihood of each ensemble member. All uncertainties are on the $1\sigma$ level, unless otherwise specified. Note that the underlying probability density function (PDF) does not have to be Gaussian or symmetric. To approximate the full PDF, we bin the quantity of interest (for example RSL), and then sum the likelihood of all ensembles that fall into each bin. The trends in EWH, deformation and RSL, together with their uncertainty estimate, are depicted in Figure 1. The regions with the largest trends and uncertainties are around the former ice sheets, while the far-field trends and uncertainties are smaller.



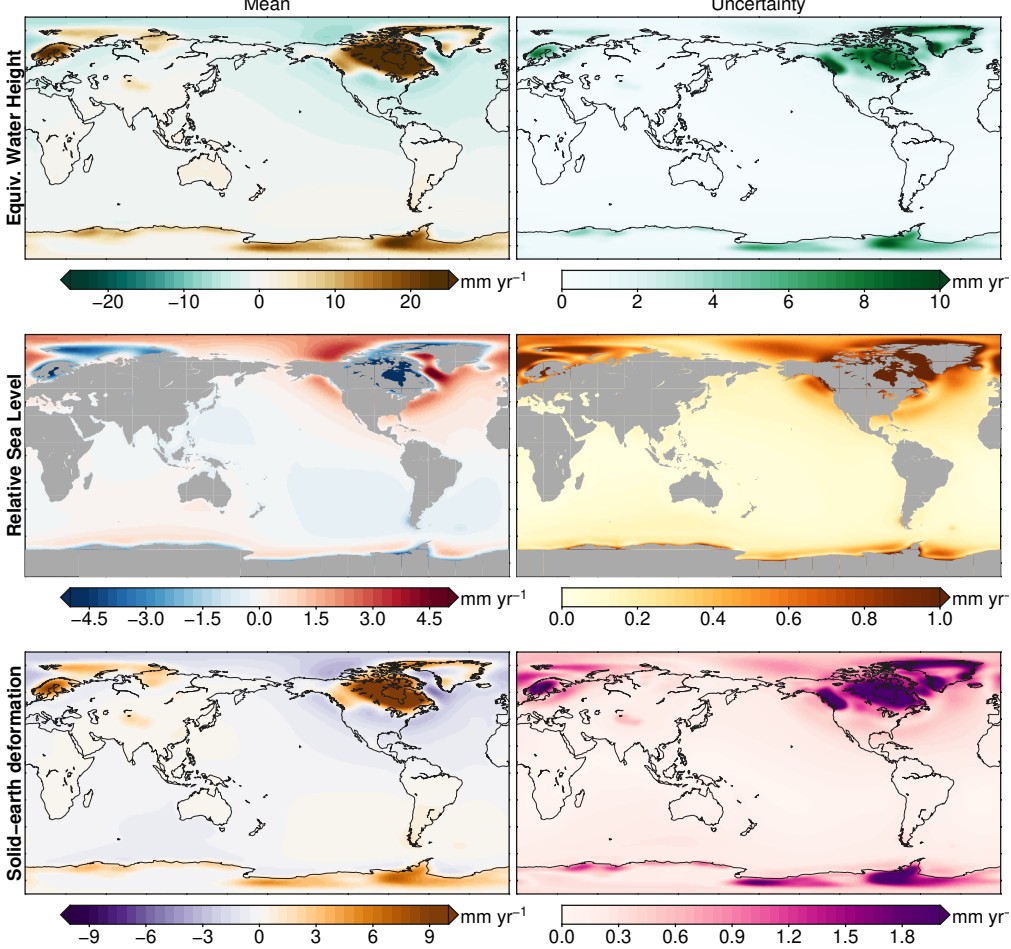

**Figure 1.** The present-day GIA trends predicted by the large ensemble. The top row shows EWH, the middle row shows RSL, and the bottom row shows deformation.

## 2.2 GRACE estimates of present-day mass redistribution

To estimate PDMR, we use the JPL GRACE Release 6 (RL06) mass concentration (mascon) solution (Watkins et al., 2015). This solution provides monthly-mean estimates of EWH anomalies from March 2002 until June 2017, with some gaps at the beginning and end of the GRACE record and has a nominal spatial resolution of 3 degrees by 3 degrees. We only look into

5  mass changes on land, and do not take ocean-bottom pressure changes driven by ocean dynamics into account. For mascons that contain a coastline, a Coastline Resolution Improvement (CRI) filter has been used to prevent the leakage of gravity signals between land and ocean (Wiese et al., 2016). The RL06 solution is corrected for geoid perturbations caused by polar wander: movements in the earth axis of rotation (Wahr et al., 2015), and comes with an estimate of the measurement uncertainty of the EWH changes in each mascon.



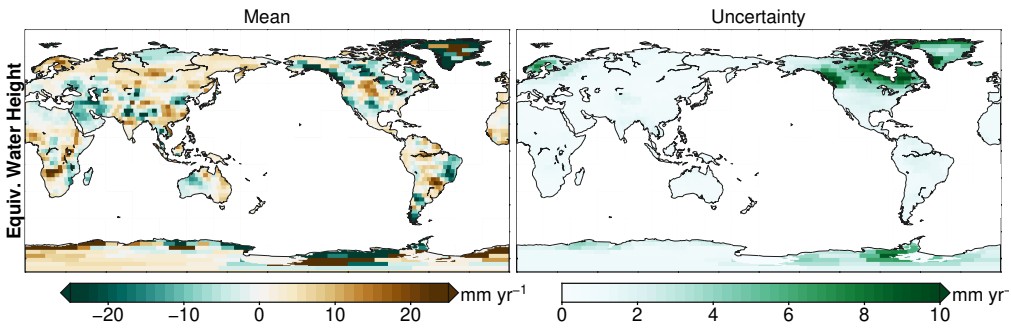

**Figure 2.** Trends and accompanying uncertainties in the EWH changes on land, observed by GRACE.

We apply the GIA correction from each ensemble member to the GRACE estimates, which yields a 5000-member ensemble of GRACE EWH estimates. Each of these realizations is then perturbed randomly using the measurment uncertainty estimate in each mascon. Following Tamisiea (2011), we adjust the degree-2, order-1 terms of the GIA model to account for the fact that GRACE observations are taken from an inertial reference frame, and not from the rotating earth. From this ensemble, we

derive the expected PDMR and the associated uncertainty, as well as the total land mass change. Figure 2 shows the trends and associated uncertainties in the GRACE estimates. The uncertainty in the trend is dominated by the GIA uncertainty, as the spatial pattern is almost identical to the pattern shown in Figure 1, and the measurement uncertainty only reaches values of about 1 mm yr$^{-1}$ EWH. Note that the uncertainty estimates of these trends are only based on the spread in the ensemble, and not on the uncertainty that arises from fitting a linear trend to the data. Since geophysical time series often exhibit serial

correlation, this assumption could result in an underestimation of the uncertainty (Williams et al., 2014).

To isolate cryospheric and hydrologic processes, we separate the observed EWH changes into changes from the Greenland Ice Sheet, the Antarctic Ice Sheet, glaciers and ice caps, and TWS. The EWH changes on both ice sheets can be isolated by only selecting the mascons that overlap with the ice sheets. In other mascons where both glaciers and TWS are potential contributors to the mass changes, a priori information is required to disentangle the components. To do so, we use the same

approach as Reager et al. (2016). First, we determine the mascons that overlap with glaciated regions, based on the Randolph Glacier Inventory (Pfeffer et al., 2014). For mascons that do not overlap with a glaciated region or ice sheet, all mass changes are attributed to TWS changes. Mass changes from the peripheral glaciers of Greenland and Antarctica are part of the mass balance of both ice sheets. For glaciers and ice caps in Alaska, Arctic Canada, Iceland, Svalbard, the Russian Arctic and the Southern Andes, we assume that the mass changes in the associated mascons are solely caused by glacier mass changes. For the

other glaciated regions, we use the pentadal glacier mass balance estimates based on geodetic and glaciological measurements from the GMBAL dataset (Graham Cogley, 2009, version R1501). The total mass change in these mascons is separated into a glacier contribution from the GMBAL data set and a TWS contribution. Note that, contrary to Reager et al. (2016), we use the GMBAL estimates for the glaciers and ice caps in the high mountain Asia region, instead of the estimates from Gardner et al. (2013), since the latter only covers the period 2003-2009, which would require an extrapolation of 8 years. Figure 3 shows the

mascon geometry, as well as the mascons of the ice sheet and glacier regions.




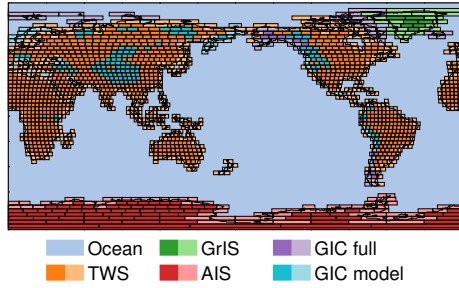

| | Ocean | | GrIS | | GIC full |
|---|---|---|---|---|---|
| | TWS | | AIS | | GIC model |

**Figure 3.** Overview of the GRACE mascon geometry and mascons associated to Greenland Ice Sheet (GrIS), Antarctic Ice Sheet (AIS), glacier and ice cap (GIC), and TWS mass changes. The light-colored mascons denote mascons which partially cover oceans and use the CRI filter. Mass changes in purple mascons (GIC full) are fully attributed to glaciers, while turquoise mascons are separated into a glacier and TWS term, see text.

## 2.3 Solid-earth deformation and sea-level changes resulting from mass redistribution

PDMR causes in the regional earth gravity field (geoid changes, $G(\theta,\phi,t)$), deformation ($R(\theta,\phi,t)$), in RSL ($\eta(\theta,\phi,t)$) and in geocentric sea level ($\zeta(\theta,\phi,t)$). $\theta$ and $\phi$ denote latitude and longitude, and $t$ time. RSL is defined as a change of the sea surface relative to the underlying sea bottom, while geocentric sea-level change is a change of the sea surface, relative to the center of

the earth. These changes have the following relationship:

$$\eta(\theta,\phi,t) = \mathcal{M}(t) + G(\theta,\phi,t) - R(\theta,\phi,t) \tag{1}$$

$$\zeta(\theta,\phi,t) = \mathcal{M}(t) + G(\theta,\phi,t) = \eta(\theta,\phi,t) + R(\theta,\phi,t). \tag{2}$$

The term $\mathcal{M}$ is needed to ensure that the total mass of the earth system is conserved. Note that this term is not necessarily equal

to the barystatic sea-level change, which is the global-mean change in $\eta$ (not $\zeta$, since changes in $R$ could make the global-mean ocean deeper or shallower) due to mass entering or leaving the ocean, and equals 1 mm for each 362 Gigaton that enters the ocean. To compute these changes, we solve the sea-level equation (Clark and Lingle, 1977), using the pseudo-spectral approach (Tamisiea et al., 2010). We assume that the solid-earth response to PDMR is purely elastic, and thus differs from viscoelastic GIA. We solve the sea-level equation in the centre-of-mass reference frame, up to degree and order 180, and include the

rotational feedback. We use the elastic love numbers from Wang et al. (2012), which are based on the Preliminary Referenced Earth Model (PREM, Dziewonski and Anderson, 1981). We solve the sea-level equation for each ensemble member, which results in 5000 estimates of deformation and RSL at each grid cell and time step.

GNSS receivers measure deformation $R(\theta,\phi,t)$ as VLM, while tide gauges measure relative sea level $\eta(\theta,\phi,t)$. Hence, tide-gauge observations that are corrected for the full VLM trend measure geocentric sea level $\zeta(\theta,\phi,t)$. This is also the

case for tide-gauge records that are corrected for VLM using the difference between altimetry and tide-gauge observations (Kleinherenbrink et al., 2018).





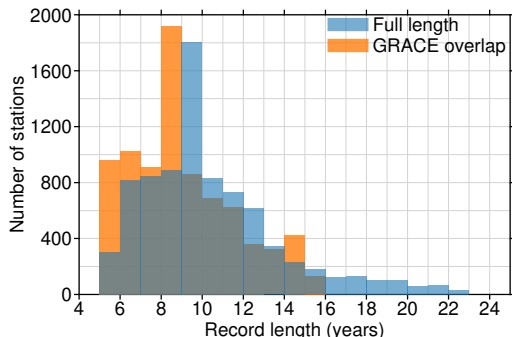

**Figure 4.** Histogram of the record length per station. Shown are all stations that fit the criteria outlined in paragraph 2.4. The blue bars show the record length before removing data outside the GRACE time span, and orange after removing.

## 2.4 GNSS stations and VLM trend estimates

We use the GNSS dataset from the University of Nevada, Reno (Blewitt et al., 2018, geodesy.unr.edu), which provides processed daily time series of over 14000 permanent GNSS receivers in the ITRF2008 reference frame. For consistency, we only use observations that overlap with the GRACE era (2002-2017). We remove stations for which less than 1825 daily observa-

tions that overlap with the GRACE era (2002-2017) are available, which corresponds to a minimum record-length of five years. This requirement, which is a trade-off between accuracy and data availability, results in a total of 8228 stations. Figure 4 shows a histogram of the record length per station, both before and after removing the observations that fall outside of the GRACE era. The histogram shows that most GNSS stations have a record length of around 8 to 10 years, or about half the length of the GRACE time span. Only a small fraction of the GNSS stations cover the full GRACE era. Since we only consider the

observations within the GRACE era, not all data can be used, which results in a decrease in the record length that is available for some stations. We interpolate the monthly VLM that results from deformation due to PDMR and GIA on the GNSS time steps, which enables us to separate the full GNSS time series into its components:

$$z_{\mathrm{obs}}(t) = R_{\mathrm{GIA}}(t) + R_{\mathrm{cryosphere}}(t) + R_{\mathrm{TWS}}(t) + z_{\mathrm{residual}}(t), \tag{3}$$

with $z_{\mathrm{obs}}(t)$ the observed height anomaly at time $t$, and $R_{\mathrm{GIA}}(t)$, $R_{\mathrm{cryosphere}}(t)$, and $R_{\mathrm{TWS}}(t)$ the uplift caused by deformation

due to GIA, ice sheets and glaciers, and TWS. $z_{\mathrm{residual}}(t)$ is the residual anomaly, which is the observed height anomaly that cannot be explained by the other terms. Combining Equation 3 with Equations 1 and 2 shows that removing the full GNSS observations $z_{\mathrm{obs}}(t)$ from the tide-gauge observations, the resulting time series denotes geocentric sea level $\zeta$.

Deformation due to GIA and PDMR could cause a net uplift or subsidence of the global and regional ocean bottom, and the total volume change of the oceans differs from the ocean volume change directly inferred from geocentric sea-level changes.

This difference results in a bias when global or regional sea-level changes are estimated from VLM-corrected tide-gauge records. Both GIA and contemporary mass redistribution result in a global-mean ocean bottom subsidence: for GIA, the global-mean subsidence is about 0.3 mm yr$^{-1}$ (Tamisiea, 2011), and for contemporary mass redistribution the subsidence is

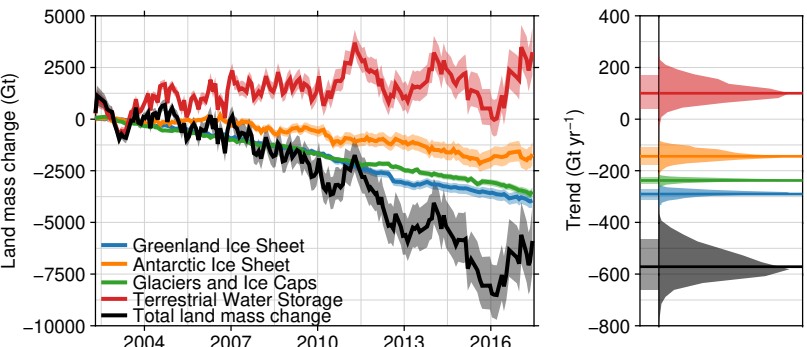

**Figure 5.** Total land mass changes as observed by GRACE, corrected for GIA. The left panel shows the expectation (thick line) and 90% confidence intervals (shade) of the mass change time series for each process and the total land mass change. Negative values denote land mass loss. The right panel shows the expected value (thick line), 90% confidence interval (left shade) and PDF (right shade) of the resulting linear trend.

in the order of 0.1 - 0.2 mm yr$^{-1}$ over the last 25 years (Frederikse et al., 2017). On regional scales, the difference between ocean-volume changes and geocentric sea-level changes can even be larger (Lickley et al., 2018). Using the trend in $z_{\text{residual}}(t)$ to correct tide-gauge records for VLM avoids this bias, since ocean-bottom deformation resulting from GIA and PDMR has been removed from the VLM records. As a result, tide gauges corrected with $z_{\text{residual}}(t)$ denote relative sea level with respect

to GIA and PDMR, while local VLM signals from other processes, such as tectonics and local subsidence are removed from the record.

   To estimate linear trends in the observed and residual height anomalies, we apply the MIDAS robust trend estimator (Blewitt et al., 2016) to the observed and residual height anomalies. The MIDAS estimator does not compute the linear trend using linear least squares, but based on the median of the height difference of each pair of anomalies that are separated by 1 year, while

the uncertainty estimate is based on the standard deviation of the 1-year separated anomaly differences. This approach has two advantages over the classical approach: 1. it is less sensitive to discontinuities in the time series, which are omnipresent in GNSS records (Gazeaux et al., 2013), and 2. is is computationally less expensive than least-squares estimation using an appropriate serially-correlated noise model. We estimate the trends for the full ensemble to obtain an uncertainty estimate for each trend. We assume that the uncertainty from the trend estimator and the ensemble spread are independent, and these terms

are added up in quadrature.

## 3   Results

### 3.1   Global-mean land-mass changes

The global-mean land-mass changes due to PDMR are shown in Figure 5, which shows that, while all cryospheric processes causes net land mass loss, the TWS term causes land mass gain. The positive TWS term has been observed over the first decade





**Table 1.** Trends in land mass changes and corresponding barystatic sea-level changes. The numbers in brackets show the uncertainties expressed as the corresponding 5-95% confidence intervals. A positive sign for the mass change correspond to increase of the mass on land, and a positive sign of the barystatic trend denotes a sea-level rise.

|  | Mass change (Gt yr$^{-1}$) | | Barystatic sea-level change (mm yr$^{-1}$) | |
|---|---|---|---|---|
| Greenland Ice Sheet | −290 | [−312 ; −272] | 0.80 | [0.75 ; 0.86] |
| Antarctic Ice Sheet | −144 | [−176 ; −108] | 0.40 | [0.30 ; 0.49] |
| Glaciers and Ice Caps | −238 | [−250 ; −227] | 0.66 | [0.63 ; 0.69] |
| Terrestrial Water Storage | 100 | [42 ; 171] | −0.28 | [−0.47 ; −0.11] |
| Total land mass change | −572 | [−659 ; −465] | 1.58 | [1.28 ; 1.82] |

of GRACE observations by Reager et al. (2016) and Rietbroek et al. (2016), and the trend continues to be positive over the full GRACE record. For all cryospheric processes, the total mass changes are dominated by the trend, and the variability around the longer-term trend is relatively small, while for TWS, the inter-annual variability around the mean trend is substantial, especially after 2010. It is known that TWS exhibits substantial decadal and multi-decadal variability (Humphrey et al., 2017; Wada et al., 2017), and a large part of this variability corresponds to the strong ENSO cycles during this period, which are known to have a large effect on water stored on land (Boening et al., 2012; Fasullo et al., 2013).

The uncertainties in the trend and time series for Greenland and glaciers are smaller than the uncertainties for Antarctica and TWS. This difference is caused by the uncertainty in the GIA contribution, which is small for Greenland and most glaciated regions, and larger and more uncertain for the Antarctic Ice Sheet. The uncertainty in the TWS term is largely caused by the uncertainty in the GIA contribution from the former Laurentide Ice Sheet, which covered large parts of North America (see Figure 2). Due to this uncertainty, the partitioning of the observed EWH changes over North America between GIA and present-day changes is uncertain, which leads to a large spread in the possible TWS contribution from this region. The total land mass mass loss over the second half of the record is larger than the loss over the first half, which is caused by the accelerating contributions from Greenland and Antarctica, and the slowdown of the TWS contribution.

The linear trends in total land mass change and the individual components are shown in Table 1, which also contains the resulting barystatic sea-level trends. The total land mass trend over the GRACE era is negative. TWS is the smallest term, but has the largest uncertainty. The distributions shown in Figure 5 are not symmetric, which causes slightly skewed confidence intervals in Table 1. The trends in the individual terms are consistent with recent studies (Bamber et al., 2018; The IMBIE team, 2018; Reager et al., 2016).

## 3.2 Regional patterns in sea level and solid-earth deformation

As discussed in Section 2.3, the mass changes will lead to regional RSL and deformation patterns. Figure 6 shows the trends in regional RSL resulting from TWS and cryospheric processes, together with the accompanying uncertainty. As expected from the barystatic contribution (Figure 5), the trends in sea level are dominated by cryospheric processes, while the TWS-induced trend in sea level is generally no more than a few tenths of millimetres per year and has a less-pronounced regional pattern.





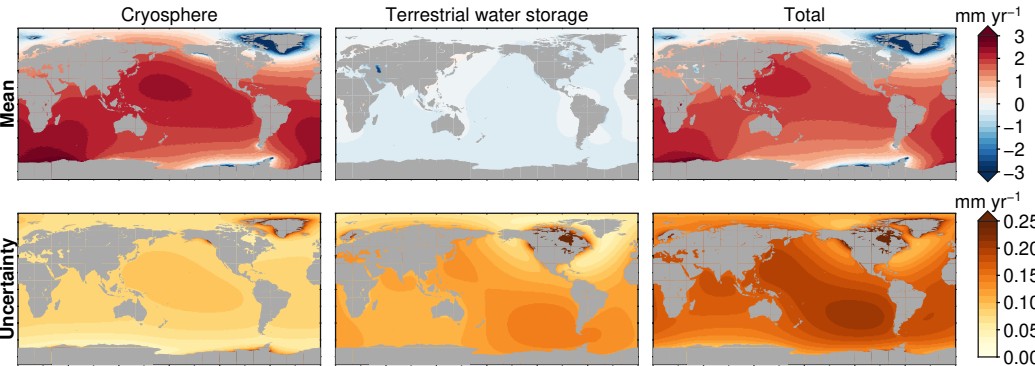

**Figure 6.** Trends in relative sea level resulting from mass redistribution observed by GRACE, separated into the cryosphere contribution (sum of glacier and ice sheet contribution), the TWS contribution, and the total land mass contribution. The trends have been computed over the full GRACE time span (2002-2017). The top row shows the mean trend, and the bottom row shows the uncertainty.

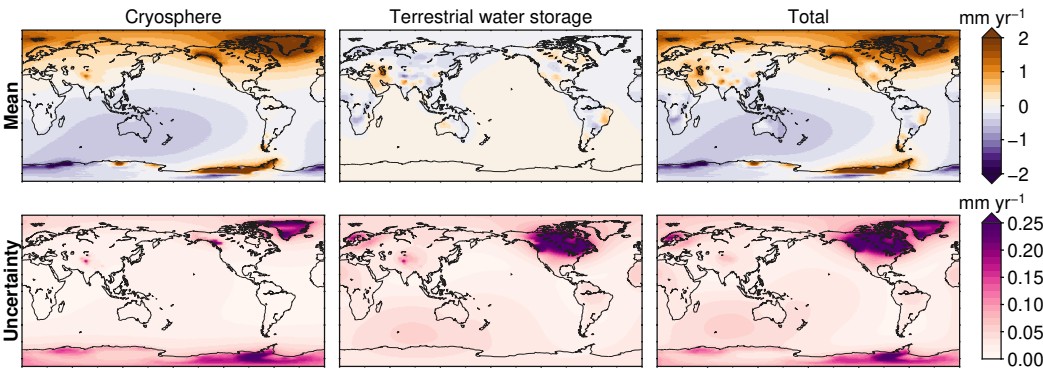

**Figure 7.** Trends in deformation resulting from mass redistribution observed by GRACE, separated into the cryosphere contribution (sum of glacier and ice sheet contribution), the TWS contribution, and the total land mass contribution. The trends have been computed over the full GRACE time span (2002-2017). The top row shows the mean trend, and the bottom row shows the uncertainty.

However, the TWS contribution shows substantial variability around the long-term trend, both for the barystatic contribution, as well as for local changes. The uncertainty in these fingerprints is dominated by the TWS contribution, and for large parts of the ocean, the uncertainty in the TWS contribution has a similar magnitude as the trend. The uncertainty for the cryosphere-induced local sea-level changes is limited to about 0.1 mm yr$^{-1}$, except for regions very close to glaciers and the ice sheets.

5    The uncertainty of the total signal is again on the order of 0.1 - 0.2 mm yr$^{-1}$, which is substantially smaller than the signal itself.

The regional deformation trends are depicted in Figure 7. Next to uplift at locations where the ice-mass loss takes place, mass changes in the cryosphere result in considerable far-field deformation signals, and causes subsidence of about 0.5 mm yr$^{-1}$ in Australia, and uplift in large parts of Europe and North America. TWS changes cause considerable near-field deformation





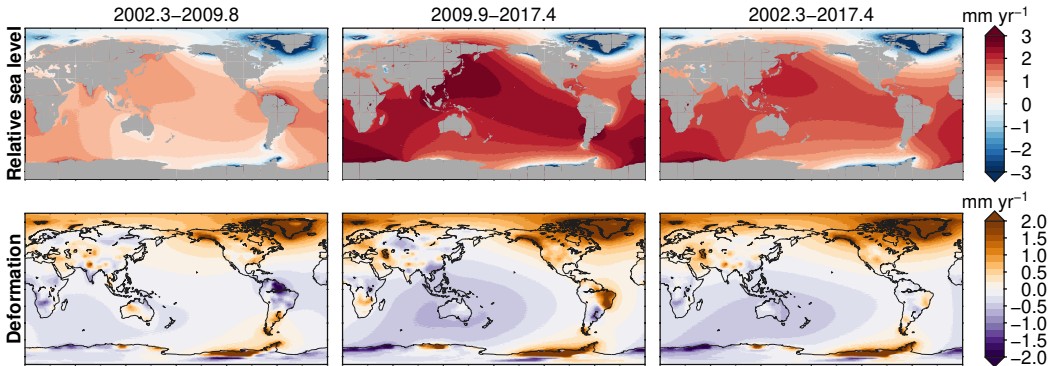

**Figure 8.** Trends in RSL (top row) and deformation (bottom row) resulting from the total mass redistribution observed by GRACE over the first (left) and second half (center) of the GRACE observation, and the total GRACE period (right). Note that deformation related to GIA is not included here.

trends, for example in South America and Asia. Both uplift and subsidence occur, which shows that the barystatic trend consists of the sum of regions of land mass loss and land mass gain, and as such, the TWS-induced deformation signals, which have a predominantly near-field signature, will likely not follow the global-mean variability. The deformation uncertainty is large close to former glaciated regions due to the local impact of GIA. For other regions, the uncertainty is below the 0.1 mm yr$^{-1}$

level, also for the locations for which the local trend is large.

The TWS-induced total land mass change shows substantial temporal variability, and the mass loss at both the Greenland and Antarctic ice sheets is accelerating. Therefore, linear RSL and deformation trends, derived over a subset of the GRACE period are likely to deviate from the trends derived over the full period. As an example of the size of these deviations, Figure 8 shows the RSL and deformation trends over the first and second half of the GRACE era. As a result of this increasing barystatic

sea level trends, the regional RSL trends are overall larger in amplitude during the second half of the GRACE era, although the spatial pattern only shows limited changes between the different periods. In contrast, local deformation trends, mostly associated with TWS changes, show vast differences between both periods. Notable regions include the Amazon basin and southern Africa, who subside over the first half of the period and show uplift over the second half of the GRACE period. The opposite occurs in the Rio de la Plata basin and Northwest Australia. In these regions, the deformation measured by the GNSS

receivers depends substantially on the time span of the record and as a result, nearby stations could see different trend when they cover different periods.

## 3.3    The role of solid-earth deformation in observed VLM trends

As discussed in the previous section, the deformation trend depends substantially on the time span of the GNSS observation. To quantify the role of deformation, we computed the trend over the time span of each GNSS station (Figure 9).

The cryosphere-driven deformation trends mostly show smooth temporal and spatial variations, which suggests that the specific time span of the GNSS record has a limited impact on the observed deformation rate. Cryosphere-induced deformation





**Figure 9.** The modelled linear trends due to deformation computed over the time span of the individual GNSS record. The left panels show the mean trend, and the right panels the uncertainty from the large ensemble. Note that the color scales for the top three rows differs from the color scale of the bottom two rows.



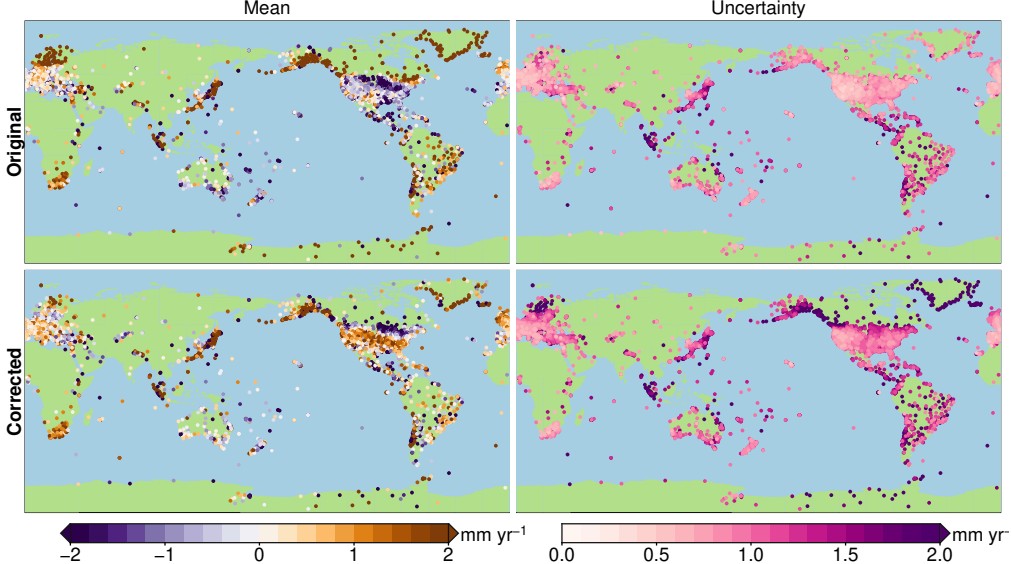

**Figure 10.** Observed VLM trends from permanent GNSS stations, together with the uncertainty. Top row: the original time series. Bottom row: time series corrected for modelled deformation due to GIA and PDMR. Both trends have been computed over the time spans for which GNSS and GRACE observations overlap.

results in substantial trends at many GNSS stations: not only the well-known near-field uplift signals, which dominate the VLM signal for many regions where ice mass loss occurs, but also in the far field, with notable uplift in large parts of Europe and the US, and subsidence in Australia. The uncertainty of these trends is negligibly small, except for the stations close to the locations of ice mass loss.

In contrast, the TWS-induced deformation trends show a less smooth pattern: sharp contrasts in the trends between nearby stations can for example be seen in North and South America, owing to the aforementioned difference in the time span of the GNSS records and the TWS trends that depend on the period over which these trends are computed. GNSS stations along the coastlines generally witness smaller deformation trends related to TWS changes, because these stations are located along the edges of the TWS load change regions, instead of in the middle.

Nevertheless, deformation trends at coastal locations can be substantial, which points at possible biases when VLM estimates used to correct tide-gauge observations do not account for TWS-induced deformation. For many regions, the deformation trend is dominated by GIA, especially for the Northeastern parts of North America and Northern Europe, while for South America and Australia, PDMR is the dominant deformation component. Not only the deformation trends, but also the associated uncertainties are mostly driven by the GIA. Outside regions with a notable GIA signal, the uncertainties in the trends are small.

When we apply Equation 3 to the GNSS observations, we can remove deformation due to PDMR and GIA from the GNSS observations. The original and residual GNSS trends are shown in Figure 10. The trends in GNSS records do show substantial spatial variability, even for nearby stations, although the uncertainty, which is generally substantial due to the short GNSS





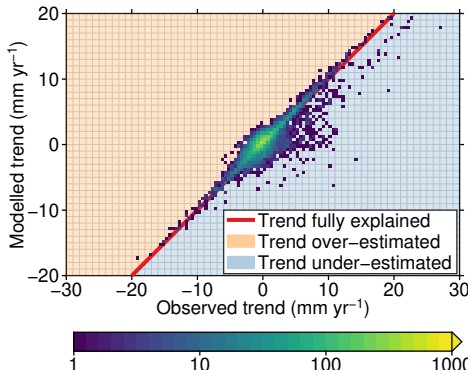

**Figure 11.** Scatter plot of observed versus modelled trends in vertical land motion. The color denotes the number of stations for each trend pair within 0.5 mm yr$^{-1}$. For points on the red line, the modelled and observed trend agree completely, in the orange area, the modelled trend is larger than the observed trend, and for the blue area, the modelled trend is smaller than the observed trend.

records, noisy data, and the presence of jumps in the data (Williams, 2003), should be taken into account when comparing nearby stations. Nevertheless, the uncorrected GNSS trends show many well-known large-scale features, mostly associated with GIA and uplift associated with present-day ice mass loss. The removal of all modelled deformation signals (Bottom panel in Figure 10) highlights regional differences: for Europe, Australia and South America, deformation explains a large part of
the observed large-scale VLM features, while for North America, a different pattern emerges, with an uplift signal over large parts of the United States. This uplift could have its cause in the GIA correction: the ensemble predicts a substantial subsidence pattern over large parts of North America, associated with the collapse of the Laurentide forebulge (Figure 1), which is stronger than projected from some other models, such as ICE6G-VM5a (Peltier et al., 2015), even considering the uncertainty. For most cryospheric regions, the trends change, but a substantial residual signal remains. A possible explanation for this large residual
is the fact that both the GIA model and the GRACE mascon solution provide PDMR estimates at relatively coarse resolutions, while it is known in these regions that deformation induced by GIA and present-day ice mass loss can have a very local character in these regions due to localized mass changes and complex earth structures (e.g. Khan et al., 2010; Hay et al., 2017). Some large-scale VLM features visible in the GNSS trends cannot be explained by the modelled deformation. For some regions, such as Japan and Chile, tectonic activity is a likely candidate, but the uplift in Southern Africa is unlikely to be tectonic in
nature. Nevertheless, in general, the model explains a substantial part of the observed vertical land motion: the scatter plot in Figure 11 shows that the modelled trend for most stations is close to the observed trend, although there are multiple stations that show large trends that are not explained by the model. Given the fact that many relevant processes, such as tectonics and local subsidence are not modelled, these outliers are not surprising.

### 3.4 Solid-earth deformation and long tide-gauge records

To determine the effects of vertical land motion on trends from long-term tide-gauge records, we partially repeat the analysis of Thompson et al. (2016, T16), which determines trends from a set of 15 long-term tide-gauge records. Since some of the stations





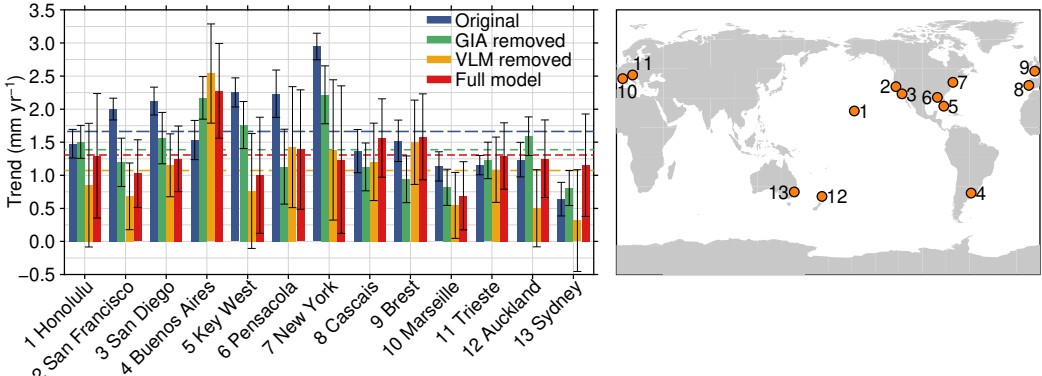

**Figure 12.** Linear trends in long tide-gauge records (1901-2000), with various corrections applied for vertical land motion. Blue: no correction, green: GIA RSL removed, yellow: original GNSS trend removed, red: GIA and corrected GNSS trend removed. The dashed lines show the median trend for each correction. The right pane shows the locations of the tide gauges. The black bars denote the $1\,\sigma$ uncertainties

from that study are not in the vincinity of a GNSS station with a long record (Cristobal), or the combination of VLM and the tide-gauge trend results in an unrealistically low sea-level trend (Newlyn, Fremantle), we have removed these stations, and added nearby stations as a replacement, where possible. See the supporting information for a complete overview of changed stations.

We apply three different corrections to the tide-gauge trends. In the 'GIA removed' model, the GIA RSL trend from the Caron et al. (2018) model is removed from the tide-gauge trend. For the 'VLM removed' model, the uncorrected GNSS trend is removed from the tide-gauge trend. For the 'full' model, we remove both the deformation-corrected GNSS trend, as well as the modelled GIA RSL trend. The sea-level trends have been computed from annual sea-level data obtained from the Permanent Service for Mean Sea Level (PSMSL, 2018; Holgate et al., 2013) using the Hector software (Bos et al., 2013),
assuming a power-law spectrum. The uncertainties due to the VLM and GIA corrections are derived from the large ensemble, and subsequently added in quadrature to the tide-gauge uncertainties. To stay consistent with T16, we only use annual tide-gauge data from 1901-2000. Figure 12 shows the uncorrected and corrected trends for the stations, as well as their location.

From the individual trends, we computed the mean and the standard deviation, which are shown in Table 2. As a comparison,
we also computed the trends for the 'GIA removed' and 'Full' models using the ICE6G-D VM5a model(Peltier et al., 2015, 2018), which is an updated version of the ICE6G-C VM5a GIA model used in T16. The 'GIA removed' and 'VLM removed' trends both have a lower standard deviation than the uncorrected trends. This reduction due to the GIA correction was also found by T16. Removing the uncorrected VLM trends from the tide-gauge trends also reduces the inter-station standard deviation, although to a lesser extent than the 'GIA removed' model. However, this correction suffers from the two aforementioned
problems: the linear trend due to deformation over the GNSS record is not representative for the full tide gauge record and the





**Table 2.** Mean and standard deviation of the trends from the long tide-gauge records shown in Figure 12 for each VLM correction. The standard deviation is calculated between the individual stations. The mean and standard deviation using the ICE6G-D VM5a model to correct for GIA are also listed for comparison.

|  | Mean (mm yr$^{-1}$) | St. dev. (mm yr$^{-1}$) |
|---|---|---|
| Original | 1.66 | 0.59 |
| GIA removed | 1.39 | 0.45 |
| VLM removed | 1.07 | 0.56 |
| Full model | 1.31 | 0.36 |
| GIA removed (ICE6G) | 1.54 | 0.35 |
| Full model (ICE6G) | 1.36 | 0.39 |

ocean-bottom deformation signal is removed from the tide-gauge data. These issues are both resolved using the 'full model', and it even further reduces the spread between stations.

Interestingly, the correction from the GIA ensemble reduces the mean sea-level trend at the tide-gauge locations by $\sim 0.3$ mm yr$^{-1}$, which is more than the $\sim 0.1$ mm yr$^{-1}$ from the correction based on the ICE6G-D model and the value found by T16. The removal of the uncorrected VLM trends results in a lower trend than the GIA-only correction. The full model results
in a trend in beween the GIA-corrected and GNSS-corrected models, but overall with the lowest standard deviation of the four models. The mean trend of the full model is close to the trend found by Hay et al. (2015) and Dangendorf et al. (2017), who find a mean trend of 1.26 mm yr$^{-1}$ and 1.21 mm yr$^{-1}$ respectively over the 20th century. Note that we do not take into account the spatial sampling bias that causes the tide-gauge records discussed here to under-estimate the underlying 20th-century GMSL
trend, as discussed in T16, so the gap discussed by T16 cannot be considered fully closed from these results. Also note that the 'full model' using ICE6G-D for the GIA correction, instead of the Caron et al. (2018) model, also results in a closer agreement between the long records and sea-level reconstructions.

## 4   Conclusions

We have quantified the effects of PDMR on relative sea-level change and VLM observations. The large ensemble of GIA
predictions allows us to quantify the uncertainty that arises from this correction, and combined with the measurement uncertainties, we have derived estimates of RSL and deformation fingerprints, together with uncertainty estimates based on mass redistribution observed from GRACE.

This approach also produces estimates of the total land-mass change, which, due to mass conservation, result in the opposite total mass change in the ocean. We find a total mass trend over the GRACE period between -659 and -465 Gt yr$^{-1}$, which
corresponds to a mass-driven sea-level trend between 1.28 and 1.82 mm yr$^{-1}$ (90% C.I.). This number is in disagreement with some other estimates of the total land mass change: WCRP Global Sea Level Budget Group (2018) find a mean estimate of 2.3 mm yr$^{-1}$ from various methods to estimate GRACE mass changes. In our ensemble, no single realization can be reconciled



with a mass change of 2.3 mm yr$^{-1}$. When evaluated over 2002-2014, we obtain a mean land mass trend of 1.4 mm yr$^{-1}$, which is consistent with the inversion approach of Rietbroek et al. (2016), who find a mass trend of $1.1 \pm 0.3$ mm yr$^{-1}$ over 2002-2014. Our total mass trend is also consistent with estimates of the individual contributors, although the error bars remain substantial. This large spread between individual estimates has consequences for the sea-level budget, and a study to reconcile these differences would be a worthwhile exercise.

The ensemble also allows for determining uncertainties for the resulting regional patterns of RSL and deformation trends. For the cryospheric processes, the trends in the regional patterns are an order of magnitude larger than the uncertainties, except for the formerly glaciated regions, where the uncertainty in the local GIA prediction plays a large role. For TWS, the uncertainties in relative sea level are generally of a similar magnitude as the trend. However, TWS changes cause substantial regional deformation trends, especially when computed over a subset of the whole GRACE time span. The deformation trends computed over the GNSS record time spans can reach values of more than 1 mm yr$^{-1}$, not only near melting ice sheets, but also in regions where large TWS changes occur, such as the Amazon Basin.

However, this approach comes with some limitations. The GRACE solutions have a limited temporal and spatial resolution. Since the mission only covers the period 2002-2017, GNSS observations from outside the GRACE period has been discarded, which, for some stations, substantially shortens the records, as displayed in Figure 4. This limitation means that the results of this study could be improved if deformation estimates are expanded to cover the full GNSS record. While for ice mass changes model results show good agreements with observations, estimating TWS changes, an important deformation source, remains challenging, which in turn limits the ability to use models to estimate deformation. Due to the coarse spatial resolution, sharp gradients in mass redistribution are smeared out over larger areas. Since deformation is sensitive to these local mass changes, the corrections computed here may under-estimate the local deformation in regions with strong spatial gradients. This issue could be one of the reasons of the un-explained residual land motion around Greenland, Antarctica, and Alaska visible in Figure 10.

Nevertheless, deformation resulting from GIA and PDMR explains a substantial part of the observed GNSS trends in multiple regions, including South America, Australia, and Europe. In contrast, we note that for some regions, such as North America, the removal results in substantial residual trends. A likely candidate for this residual trend is the uncertain GIA contribution. Since we use a global model, which is not optimized for a specific region, regional misfits may occur. Furthermore, local and regional VLM may find its origin in other processes than deformation driven by GIA and PDMR.

Because of PDMR-driven deformation, VLM trends derived over the GNSS record length can be substantially affected by PDMR variability, which causes biases when these trends are extrapolated to explain VLM over longer tide-gauge records. This bias could affect global and regional sea-level reconstructions and projections that depend on VLM-corrected tide gauges. VLM corrections derived from differences between altimetry and tide-gauge observations (Wöppelmann and Marcos, 2016; Kleinherenbrink et al., 2018) are also affected by this bias. Correcting tide-gauge observations with the corrected trends instead of original trends avoids this bias.

For a set of long tide-gauge records, correcting tide gauges for the deformation-corrected GNSS trends not only results in a smaller spread between stations, but it also reduces the gap between the long-term trend at these stations and trends





found in recent global sea-level reconstructions. However, in this study we haven't fully separated the observed trends: many unmodelled processes remain in the data, and a full understanding of these processes is key to understand the impact of vertical land motion on tide-gauge observations. We hope that the method presented here will serve as a base for future studies to further separate the observed VLM trends into individual components by integrating new models of physical processes, such

as the deformation due to post-seismic relaxation, sediment compaction, and groundwater depletion.

*Data availability.* The GRACE estimates, deformation and RSL fields, and the original and corrected trends can be downloaded from https: //thomasfrederikse.stackstorage.com/s/SUgiCU24voIR6j4. The data and codes have been obtained from the following web sites: GNSS time series and MIDAS code: http://geodesy.unr.edu/, the GMBANAL R1501 glacier mass balance estimates: http://people.trentu.ca/~gcogley/ glaciology/glglmban.htm. RGI glacier region masks: http://www.glims.org/RGI/. Tide-gauge observations:http://www.psmsl.org. ICE6G-

D GIA model predictions: http://www.atmosp.physics.utoronto.ca/~peltier/data.php. Hector trend estimation software: http://segal.ubi.pt/ hector/

*Author contributions.* TF and FWL conceived the idea for the study, TF and LC ran the computations, and all authors contributed to the manuscript.

*Competing interests.* The authors declare no competing interests

*Acknowledgements.* All figures have been made using the Generic Mapping Tools (GMT). This research was conducted at the Jet Propulsion Laboratory, which is operated for NASA under contract with the California Institute of Technology.



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
