# Peer review of "The imprints of contemporary mass redistribution on local sea level and vertical land motion observations"

_Solid Earth, 2018_

## Referee Comment (RC1) · Anonymous Referee #1 · 19 Feb 2019

Summary: The manuscript presents an original approach to estimate trends in vertical land motions (VLM) and relative sea levels (RSL) due to the present-day mass redistribution (PDMR) occurring during the GRACE (Gravity Recovery And Climate Experiment) satellite mission (March 2002 – June 2017). PDMR changes are evaluated using the JPL (Jet Propulsion laboratory) GRACE mascon solutions [Watkins et al., 2015] and separated into a cryospheric and terrestrial water storage components, depending on the geographical location of mascons. This analysis is completed with an estimation of VLM and RSL trends due to Glacial Isostatic Adjustement (GIA), using the ensemble of GIA models from Caron et al. [2018]. The predicted VLM trends due to changes in the cryosphere, TWS and GIA are then compared to observations,

using a subset of the GNSS data from the University of Nevada [Blewitt et al., 2018] matching the GRACE observation period. Finally, the authors discuss the impact of different VLM corrections on secular sea level rise averaged using a subset of 13 long tide gauge records.

Major comment: The manuscript provides a robust analysis of RSL and VLM trends due present-day ice melting, terrestrial water storage changes and glacial isostatic adjustment during the GRACE era. The link towards longer time scales is however not convincing. The major issue associated with the extrapolation of VLM corrections on secular time-scales is that we only have a very limited observation window (Figure 4, p7) and strongly non-linear processes. The problem is well stated by the authors (e.g. L10-12 p1), but, for several reasons, I doubt that their approach is appropriate to answer the issue, as it is claimed throughout the manuscript (e.g. L13-16 p1, L32-34 p2, L1-2 p16, L30-34 p17).

(i) VLM observations are decomposed in a cryosphere, TWS, GIA and residual components (eq. 3 p7), which except for GIA (and even this is arguable), all mix linear and non-linear processes. Therefore, non-linear processes in the residual VLM, due to local groundwater depletion, seismic deformation or other processes (e.g. L23-24 p2), are still present in the VLM correction and will generate a bias when extrapolated at longer time-scales.

(ii) PDMR observations cannot fully account for the non-linear processes related to the cryosphere and TWS, given the limited time-span and spatial resolution of GRACE observations. These unmodelled processes will also end up in the residual VLM, be extrapolated at secular time-scales, and bias the correction applied to tide gauge observations.

(iii) The authors have tested their approach on a very limited subset of tide gauges, comprising only 13 sites. It is difficult to believe that this sample is statistically significant. RSL changes arise due to a complex mix of processes, that can be easily aliased

with linear or non-linear VLM on such a small subset. The PSMSL database comprise many tide-gauge records that are significantly longer than VLM observations (with at least 370 tide gauges with more than 50 years of observations), that can be used to test the validity of VLM corrections. Besides, given that the main issue here is the extrapolation of non-linear processes in time, it would be good to show the impact of these corrections in a time-series analysis. Sole the mean and standard deviation of RSL trends are provided here, which is insufficient to assess the temporal and spatial variability expected from tide gauge measurements.

Further analysis is therefore required to validate the assumption of the authors (i.e. their approach allows to get rid of the bias when extrapolating a VLM correction, based on the analysis of GNSS data, on secular time-scales). The paper could be limited to the analysis of recent (March 2002 – June 2017) mass exchanges between the continents and the oceans and their impacts on barystatic sea level changes, vertical land motions and relative sea level changes. The results would have to be put in context (the tool developed is not adapted to solve the issue brought forward here) and bring some clarifications on the following points.

Other comments and questions:

General comment on methodological aspects: The "Data and methods" section is rather difficult to read. It would help to have a small paragraph and/or flowchart giving an overview of the method, that would link the various observational datasets and modelling approaches that are used together. The equation 3 at p7 is quite helpful to understand the author's approach but should come sooner in the paper.

Section 2.1 GIA model

p3: Is the ensemble of GIA predictions extracted from Caron et al [2018] only applied as an a-posteriori correction to the GRACE mascons solutions? Please, confirm or correct in the manuscript.

Section 2.2 GRACE etc.

p5 L3-4: "Each ... mascon" How? What noise model is used? (+ typo in measurement)

p5 L6: "The uncertainty in the trend is dominated by GIA uncertainty": What are the other sources of uncertainty and how are they estimated?

p5 L19-21: Can you clarify how the separation between the cryosphere and TWS is done in mixed mascons?

Section 2.3: Solid earth deformation etc.

p6"we solve the SLE using the pseudo-spectral approach (Tamiesia, 2010)": If I'm correct, this requires to express the load in spherical harmonics. How was this done? At which order? Please, also precise what load model is used (GRACE-derived PDMR mascons?)

Section 2.4: GNSS etc

p7 L7-9: This stresses the issue of the record length, which is too short to account for non-linear changes in VLM, and extrapolate them on longer time-scales.

p8 L2-6: "Using the trend in zresidual (t) ... from the record": I do not understand the logic here. Once again, z residual is not supposed to be linear in any way, it is probably the largest source of error in the extrapolation of VLM correction at longer time-scales.

Section 3: Results

p8 last line: How do you explain a net land mass gain in TWS?

You do mention later in the conclusion (p16 L20-22) that the global sea level rise due to PDMR (1.58 mm/yr Table 1) disagree with other estimates, that are usually higher. Why is the cryosphere contribution smaller with your approach?

p 11 L6-7: Can you quantify the acceleration terms in ice mass loss? Is it comparable with other estimates?

p 11 L20 : What is your indicator of "smooth temporal variations"?

p 13 L9 : Because GRACE resolution and mascon geometry is not adapted near the coast? Following that train of thought, it is unlikely that GRACE-derived TWS changes can account for the strong local variability evidenced at tide gauges.

p 14 L10: this can easily be estimated with the application of a spatial filter on the observations (e.g. Pfeffer et al., 2017).

p 14 L15 to 18: can you provide some quantitative information about the agreement/disagreement between VLM observations and predictions (root mean square error, coefficient of determination, bias, maximum differences, distribution of the differences etc.)

p14 L20: why do you limit your analysis to these 15 stations? why such a restrictive selection?

p15 L7 to 8 "For the full model . . . GIA RSL trend" Why? How does that help with the extrapolation of non-linear processes in time?

p 15 L10 "assuming a power law spectrum" to describe temporally correlated noise?

p 15 L11: Why do you choose the T16 subsample? How does it help to validate your approach?

p 16 L1-2: How are these issues resolved?

p16 L9: Yes, the "sampling bias" is a huge problem that needs to be addressed. More observations are available to start with.

Section 4: Conclusion

p 16 L20: Why this disagreement? This should be discussed earlier in the manuscript.

p 17 L6: How do you estimated uncertainties that are not related to GIA?

p 17 L 23: please provide metrics to illustrate the agreement/disagreement between

VLM observations and predictions.

p 17 L 33: The authors did not convince me that their approach avoids the bias, due to the extrapolation of non-linear VLM. It does not logically follow their assumptions and has not been evidenced in the results. However, they provide insights on the mechanisms driving recent (GRACE era) sea level changes and solid earth deformation, which is, I think, useful results.

Details:

Abstract:

L1 to 3: rephrase the first sentence to be more readable (less "and", please)

L8: "the temporal variations affect GNSS-derived VLM": temporal variations of what?

L13 to 16: This is very confusing. Separating VLM observations in GIA and PDMR components does not separate linear from non-linear components (see major comment).

Introduction:

L32 to34: see major comment

Data and methods:

L10: "GIA affects GRACE observations . . .": replace by causes changes in the gravitational potential observed by GRACE satellites or equivalent to keep the same structure for each proposition of the sentence.

L27 p3: "we bin the quantity of interest": not sure what bin means

eq 3: why using two different variables (R and z)?

p 7 L14: use of the term "uplift": replace by vertical displacement of the earth's surface (or equivalent)?

Results p10 L8 and 9: "considerable" means how much?

Conclusion p16 L14- 17: please reformulate to clarify

Figures: It is difficult to read the color scales of figure 7, 8 (bottom row), 9 and 10. Is possible to provide a bit more contrast to have a better idea of the range of variations of RSL and VLM trends? Otherwise, provide numbers in the legend or in the text.

---

## Referee Comment (RC2) · Anonymous Referee #2 · 4 Jul 2019

The paper aims to develop estimates for deformation due to GIA and present-day mass redistribution with uncertainty estimates. The deformation patterns are explained and used to correct tide-gauges. It is an original idea to address a highly relevant problem. It is shown that this correction improves consistency between sea-level trends from tide gauges and sea level reconstructions, and offers potential for better regional sea level projections. I see no problems in the methodology. Therefore I recommend the paper to be published after addressing two main comments below, and the long list of textual comments connected to these.

Correcting tide gauges only holds for the time period of the data, or the time for which

[Figure]

the model is valid, while one of the reasons for decomposing the relative sea level rate is to make projections outside the data range. The corrections are necessarily based on limited time span and also do not include regional deformation processes which can vary in time. This is an important limitation of the paper that is not discussed well in section 3.4 and in the conclusions, see specific comments below.

2. A second problem in the paper is the way it is written. Logic is sometimes hard to follow, procedures are not described clearly or are not explained, the use of words that should have a precise meaning is a bit sloppy (words such as trend or linear trend, deformation, model, relative sea level or just sea level). Several examples are given below. A thorough revision of the text is necessary because it now guessing is required from the reader in several places to put the pieces of the puzzle together.

A pdf is attached with small textual comments or typos.

Specific comments

The abstract is a mix of describing the processes and the methodology. I suggest to move the methodology to the last paragraph where the methodology is now partly described

Introduction The introduction is difficult to follow because it is a mix of a background and methodology, and the description of the objective is scattered. I suggest to reorder the introduction to more clearly separate background, problem statement, methodology and application separately. Also the first part of the conclusion and the introduction should be better aligned.

Page 1 Suggest to add Wu and Peltier (1984) to Milne and Mitrovica (1998)

Page 2 - line 3 I get the impression that sea level and relative sea level are not used consistently according to their definitions, please check - line 15 and further: use trend or linear trend consistently - line 34: "to avoid this possible bias" Please make clear what bias you refer to and what part you aim to remove. If the bias refers to the

(local) processes in line 21 and further then correcting for PDMR and GIA alone is not sufficient. If this bias refers to the bias due to PDMR alone (line 28 and further) it is not clear why you would remove GIA as well. Also the time-period seems relevant because you can not use the computed PDMR induced deformation beyond the period of the data.

Page 3 - Line 6: the text contains 'estimates of GIA', 'GIA solutions','model ensemble'. Please make it more consistent. - Line 9: title of section 2.1 , I suggest to use something like 'prediction ensemble' because you do not actually discuss the model - line 15: It seems to be partly circular reasoning when you use model ensembles scored according to fit to GNSS data to correct GNSS data. Please address this in the text.

Page 4 - Line 5: please explain why ocean bottom pressure changes are not used, because they are a form of PDMR - Line 6: please discuss the possible effect, if any, of this filter on the final estimates, as the main interest is in deformation along the coast.

Page 5 Potential uncertainty from determining deformation due to ice sheets, glaciers and TWS should be addressed here

Page 6 - Line 13: It is more useful to say what is neglected: viscous effects due to PDMR changes before and during the period of interested, and where these effects could occur - Line 15: please add a reference as there has been discussion about the methodology

Page 7 Line 14: how is the height anomaly defined?

Section 2.4 It would help the reader if you explain what result you are after in this section before describing possible corrections.

Page 9 line 7-15: There are probably existing studies on TWS and land mass changes that these results can be compared too.

page 11 section 3.2, the last paragraph in the section needs a conclusion, which could perhaps be moved here from section 3.3 - line 18: do you mean the solid earth deformation trend from section 3.2? How do GNSS observations play a role there? - line 20: the temporal variations are not shown in figure 11

page 13 - line 3: It should be explained what kind of cryosphere changes could cause these kind of uplifts

page 14 Line 2: discussion of the uncorrected trend could be moved before the statement that the observations will be corrected.

Line 8: "even considering the uncertainty." This is ambiguous. Please make clear whether you mean that the ensemble mean is stronger or not, or whether you talk about a statistically significant increase Line 20: "partially repeat the analysis". Describe the analysis because now it is not clear what you are doing different and why in what follows.

Page 15 line 1: specify what you mean by "in the vicinity" From line 5 onwards the text is very hard to follow. You need to explain that the goal is and why certain choices are made. The only explanation is that the analysis of Thompson et al. (2016) is partially repeated. Please add intepretation of the figure to line 12, now it is left to the reader. Several comments and questions on this section can be found in the pdf.

Page 16 line 1: I don't agree that both issues are resolved (same for page 17 line 32). Regional deformation such as given in page 2 line 22 will also not follow a constant trend so you can not use GNSS data or models with a shorter period than the tide-gauge period and expect that extrapolation of deformation models or data works fine, or am I missing something? Line 10: "the gap discussed by T16" specify which gap for readers that have not read that paper

Page 17 Line 25: "uncertain GIA contribution" explain if this means that the uncertainty in the GIA models ensemble is underestimated

Please also note the supplement to this comment:

https://www.solid-earth-discuss.net/se-2018-128/se-2018-128-RC2-supplement.pdf

[Figure]

**Supplement:**

[revised manuscript text omitted]

---

## Author Comment (AC1) · 6 Aug 2019

We'd like to thank both referees for their comments, which have helped us to improve the manuscript. Next to all changes to the manuscript as a response to the remarks of the reviewers, which are all listed below, we have also made the following adjustments to the paper:

- We found a bug in our code, which caused an underestimation of the GIA correction that was applied to the GRACE observations, which led to bias mass changes. We have resolved this bug and updated all results accordingly. The updated results do not significantly alter any of the conclusions from this paper.
- We have updated the glacier mass balance dataset that was used to separate glacier mass changes from terrestrial water storage changes to a more recent version from Zemp et al. (2019).

Please find below a point-to-point response to all remarks from both referees. The referee comments are in black, our response is in blue, and updated manuscript text is in orange.

On behalf of all authors,
Thomas Frederikse

**Referee 1**

Summary: The manuscript presents an original approach to estimate trends in vertical land motions (VLM) and relative sea levels (RSL) due to the present-day mass redistribution (PDMR) occurring during the GRACE (Gravity Recovery And Climate Experiment) satellite mission (March 2002 – June 2017). PDMR changes are evaluated using the JPL (Jet Propulsion laboratory) GRACE mascon solutions [Watkins et al., 2015] and separated into a cryospheric and terrestrial water storage components, depending on the geographical location of mascons. This analysis is completed with an estimation of VLM and RSL trends due to Glacial Isostatic Adjustement (GIA), using the ensemble of GIA models from Caron et al. [2018]. The predicted VLM trends due to changes in the cryosphere, TWS and GIA are then compared to observations, using a subset of the GNSS data from the University of Nevada [Blewitt et al., 2018] matching the GRACE observation period. Finally, the authors discuss the impact of different VLM corrections on secular sea level rise averaged using a subset of 13 long tide gauge records.

**Major comment**

The manuscript provides a robust analysis of RSL and VLM trends due present-day ice melting, terrestrial water storage changes and glacial isostatic adjustment during the GRACE era. The link towards longer time scales is however not convincing. The major issue associated with the extrapolation of VLM corrections on secular time-scales is that we only have a very limited observation window (Figure 4, p7) and strongly non-linear processes. The problem is well stated by the authors (e.g. L10-12 p1), but, for several reasons, I doubt that their approach is appropriate to answer the issue, as it is claimed throughout the manuscript (e.g. L13-16 p1, L32-34 p2, L1-2 p16, L30-34 p17).

(i) VLM observations are decomposed in a cryosphere, TWS, GIA and residual components (eq. 3 p7), which except for GIA (and even this is arguable), all mix linear and non-linear processes. Therefore, non-linear processes in the residual VLM, due to local groundwater depletion, seismic deformation or other processes (e.g. L23-24 p2), are still present in the VLM correction and will generate a bias when extrapolated at longer time-scales.

We fully agree with the reviewer that our method does not fully solve the problem of non-linear processes when extrapolating the results. Our line of reasoning in this regard has been the following: the current state-of-the-art processing is just to extrapolate the trend derived from short GPS records along the whole tide-gauge record (e.g. Wöppelmann et al. 2014, Dangendorf et al. 2017), which has two limitations:

1. Vertical land motion derived over the short GNSS record is not per se representative for the long-term VLM signal.

2 When removing the VLM signal from tide-gauge records, we obtain geocentric sea level. As a result, sea-level reconstructions based on these records reconstruct global-mean geocentric sea level, which underestimates global-mean sea level because deformation of the sea floor is disregarded.

With our proposed method we remove one process that causes problem 1, while we also reduce the impact of problem 2. The reviewer is indeed right that there are many more processes that cause problem 1, which include processes we don't model (earthquakes, local subsidence from groundwater extraction etc.), and processes we do model, but with a sparse resolution inherent from the GRACE data, which could also cause unmodeled signals related to GIA and PDMD that are retained in the signals.

In the introduction, we added the following description to describe our line of reasoning, while again warning that we do not remove other processes from the GNSS data:
P1L30: In this paper, we propose an alternative approach to correct tide gauges for VLM: instead of removing the trend in observed VLM from GNSS records, the SED resulting from contemporary mass redistribution can be removed from the GNSS time series before computing the VLM trends that are used to correct tide gauges. With this method, we retain all other processes that cause local VLM, but we avoid that decadal and multi-decadal variability from contemporary mass redistribution aliases into VLM trends estimated from short GNSS records.

While not arguing that we fully solve the problem of VLM non-linearities, given the large impact of solid-earth deformation from contemporary mass redistribution, we do argue that the method we bring forward here is an improvement over the current practice of extrapolating the GNSS trends over the tide-gauge era. However, we sincerely want to avoid the unjustified impression that this work solves all problems with non-linearities in vertical land motion. Therefore, we have re-written parts of the abstract, introduction, methods, and conclusions to ensure that we do not 'oversell' this method.

In the abstract and introduction we now explicitly state that we only deal with solid-earth deformation due to GIA and contemporary mass redistribution, and we have added an explicit warning in the methods section:
P4L3: The term $R_{residual}(t)$ still contains all VLM that results from other processes. Many of these processes act on centennial time scales, such as sediment isostatic loading, compaction, and low-frequency tectonic processes. However, many other processes that act on shorter time scales, including groundwater depletion, hydrocarbon extraction, and co-seismic deformation, are also still present in the data. Therefore, extrapolating the trend in $R_{residual}(t)$ does avoid the bias due to contemporary mass redistribution, but not due to any other process that shows interannual and decadal variability, which means that the trend in $R_{residual}(t)$ does not represent the definite secular background trend.

In the conclusions we repeat this warning:
P19L30 Another important limitation is that we only consider the effects of solid-earth deformation due to GIA and contemporary mass redistribution, while many other local and large-scale processes, such as tectonics and local subsidence due to groundwater and hydrocarbon extraction, are still present in the residual VLM time series. Like SED, many of these processes are also highly non-linear, and therefore also cause problems when records are extrapolated. Therefore, the linear trend in residual VLM that we have computed cannot be regarded as the secular background trend that is free from any bias when extrapolated back in time. A full understanding of these processes is key to fully understand the impact of vertical land motion on tide-gauge observations. We hope that the method presented here will serve as a base for future studies to further separate the observed VLM trends into individual components by integrating new models of physical processes.

(ii) PDMR observations cannot fully account for the non-linear processes related to the cryosphere and TWS, given the limited time-span and spatial resolution of GRACE observations. These unmodelled processes will also end up in the residual VLM, be extrapolated at secular time-scales, and bias the correction applied to tide gauge observations.

The GRACE data indeed has a limited spatial and temporal resolution. We avoid the bias in residual VLM due to the limited time span by only using GNSS data that overlaps with GRACE, which is shown in Figure 5. We have re-visited the sections that discuss the limitations on the spatial resolution and the effects of local solid-earth properties in the conclusions section:

P19L23 Due to the coarse spatial resolution of the GRACE data, sharp gradients in mass redistribution are smeared out over larger areas. Since SED is sensitive to these local mass changes, the corrections computed here may under-estimate local SED in regions with strong spatial gradients. This issue could be one of the reasons of the un-explained residual land motion around Greenland, Antarctica, and Alaska visible in Figure 11. Another limitation is that we compute SED with an elastic model that assumes a laterally uniform Earth structure. In some regions, such as West-Antarctica, elastic properties can deviate from their global-mean values and visco-elastic effects could occur on decadal time scales, which leads to additional deformation on top of the elastic response (e.g. Hay et al., 2017).

(iii) The authors have tested their approach on a very limited subset of tide gauges, comprising only 13 sites. It is difficult to believe that this sample is statistically significant. RSL changes arise due to a complex mix of processes, that can be easily aliased with linear or non-linear VLM on such a small subset. The PSMSL database comprise many tide-gauge records that are significantly longer than VLM observations (with at least 370 tide gauges with more than 50 years of observations), that can be used to test the validity of VLM corrections. Besides, given that the main issue here is the extrapolation of non-linear processes in time, it would be good to show the impact of these corrections in a time-series analysis. Sole the mean and standard deviation of RSL trends are provided here, which is insufficient to assess the temporal and spatial variability expected from tide gauge measurements. Further analysis is therefore required to validate the assumption of the authors (i.e. their approach allows to get rid of the bias when extrapolating a VLM correction, based on the analysis of GNSS data, on secular time-scales). The paper could be limited to the analysis of recent (March 2002 – June 2017) mass exchanges between the continents and the oceans and their impacts on barystatic sea level changes, vertical land motions and relative sea level changes. The results would have to be put in context (the tool developed is not adapted to solve the issue brought forward here) and bring some clarifications on the following points.

We are afraid that we did not accurately describe the purpose of the application of our method to the long tide-gauge records, which is indeed not an appropriate validation of our method due to the low number of records. We have added this section, because is serves as an example to show that vertical land motion and contemporary mass redistribution have a large impact on trends estimated at tide-gauge locations.

The reason why we chose these tide gauges is that the discrepancy found by T16 is that this paper has shown a discrepancy between global sea-level reconstructions and long tide gauges. While it generated quite some attention in the sea-level community, the discrepancy has not yet been solved. In this example, we show that our proposed method could explain a part of this discrepancy.

We have added extra remarks to explicitly warn that this subset of tide gauges should not be seen as a validation of our method, but merely as a possible explanation of the T16 discrepancy:

P3L5: As an example of the effect of SED on VLM-corrected tide-gauge records, we revisit the analysis of Thompson et al. (2016, T16), who showed that the 20th-century sea-level trends observed at a set of long high-quality tide-gauge records could not be reconciled with the global-mean sea-level trends from recent reconstructions. We apply the residual VLM trends to the long tide-gauge records to see whether land motion could explain a part of this discrepancy.

A question that indeed remains open, is to which extent this new method affects other tide gauges. The major reason why we do not analyze all tide-gauge records in the PSMSL database is mainly because pairing tide gauges with GNSS stations is a non-trivial task, and it generally comes down to manually pairing GNSS stations with tide gauges. We'd argue that this task falls outside the scope of this paper. We have added a warning to the conclusions that the results from this subset should not be generalized to other tide gauges or global reconstructions:

P19L13: Note that we have only applied our method to a limited subset of tide gauges, which means that the reduction in local sea-level trends and the spread among stations is not necessarily representative for other tide gauges. Whether correcting tide gauges using our method affects global sea-level reconstructions is therefore still an unanswered question.

**Other comments and questions:**

General comment on methodological aspects: The "Data and methods" section is rather difficult to read. It would help to have a small paragraph and/or flowchart giving an overview of the method, that would link the various observational datasets and modelling approaches that are used together. The equation 3 at p7 is quite helpful to understand the author's approach but should come sooner in the paper.
We agree with the reviewer that the methodology section is not really clear. Reviewer 2 also brought this forward. Therefore, we have rewritten the methods section, and we have added a flowchart of the followed procedure as an extra figure, which hopefully helps in understanding and reproducing our results. The equations that describe our definition of 'Residual VLM' are now located at the beginning of the methods section.

*Section 2.1 GIA model*

p3: Is the ensemble of GIA predictions extracted from Caron et al [2018] only applied as an a-posteriori correction to the GRACE mascons solutions? Please, confirm or correct in the manuscript.
Yes, we use the ensemble to correct the GRACE mascon solution for GIA after restoring the original GIA correction applied to these mascons, but we also use the GIA ensemble to directly derive the GIA-related solid-earth deformation at the GPS locations. The new flowchart figure and the updated methods section now mention this procedure.

*Section 2.2 GRACE etc.*

p5 L3-4: "Each . . . mascon" How? What noise model is used? (+ typo in measurement)
We use the uncertainty estimates that come with the JPL mascon product, which are based on the error covariance matrix of the solution. Since the mass change estimates of in each mascon are more or less independent from its neighbors and other time steps, we assume that these errors are uncorrelated between time steps and mascons. The noise model is therefore 'white'. We have added an explanation to this section to clarify this.

p5 L6: "The uncertainty in the trend is dominated by GIA uncertainty": What are the other sources of uncertainty and how are they estimated?
We consider uncertainties in the measurement and in the GIA model, see also the answer to the previous remark. We have added a remark to clarify this.

p5 L19-21: Can you clarify how the separation between the cryosphere and TWS is done in mixed mascons?
Yes, for these mascons that contain glaciers, but which are not dominated by glaciers, we do the following:
We use the in-situ observations from Zemp et al. (2019) to estimate the glacier mass loss in these mascons. Then, we can compute the TWS mass loss by subtracting the in-situ glacier mass loss from the total mass change in the mascon, as observed by GRACE. The in-situ mass changes from Zemp et al. (2019) come with an uncertainty. To propagate this uncertainty into the final estimates, we perturb each ensemble member based on this uncertainty. There was a short note on this in Paragraph 2.2, but we have extended the explanation to avoid any confusion.

*Section 2.3: Solid earth deformation etc.*

p6 "we solve the SLE using the pseudo-spectral approach (Tamiesia, 2010)": If I'm correct, this requires to express the load in spherical harmonics. How was this done? At which order? Please, also precise what load model is used (GRACE-derived PDMR mascons?)

Yes, that is indeed correct. The pseudo-spectral approach requires multiple synthesis and analysis steps between spherical and spherical harmonic coordinates, which we do up to degree and order 180 using the shtns library (Schaeffer et al. 2013). The input load is indeed the GRACE-derived PDMR mascon solution. We subsequently solve the sea-level equation for each ensemble member. We have revised the explanation to make this procedure clear.

*Section 2.4: GNSS etc*

p7 L7-9: This stresses the issue of the record length, which is too short to account for non-linear changes in VLM, and extrapolate them on longer time-scales.

Yes, we fully agree. As discussed before, this problem is one of the main reasons to write this paper. We have changed the introduction section to highlight this.

p8 L2-6: "Using the trend in zresidual (t) . . . from the record": I do not understand the logic here. Once again, z residual is not supposed to be linear in any way, it is probably the largest source of error in the extrapolation of VLM correction at longer time-scales.

We are indeed far from sure (up to the point that we are quite sure of the opposite) that the residual VLM term is linear and can be safely extrapolated. However, with this procedure, we have at least removed one term that is not linear from the VLM data. See also our response above.

*Section 3: Results*

p8 last line: How do you explain a net land mass gain in TWS? You do mention later in the conclusion (p16 L20-22) that the global sea level rise due to PDMR (1.58 mm/yr Table 1) disagree with other estimates, that are usually higher. Why is the cryosphere contribution smaller with your approach? p 11 L6-7: Can you quantify the acceleration terms in ice mass loss? Is it comparable with other estimates?

During the revision process, we found a bug in the GIA correction that we applied to the GRACE data. We have resolved this bug, and as a result, the net land mass gain in TWS is not significant anymore, and the mass change estimates that we find are in line with most other estimates. We have updated the text, tables, and figures to reflect this. Furthermore, as explained above, we have decided to remove the discussion about our mass trends from the conclusions, as computing barystatic trends is not the main purpose of this paper. Although quantifying the acceleration in all these terms would be a very interesting study, we have decided not to include it in this manuscript for the following two reasons:

1. We want to limit the scope of this paper on how solid-earth deformation as derived by GRACE observations affect VLM at GNSS and tide-gauge locations. For this purpose, we have to come up with mass change estimates, but updating mass change estimates from glaciers, ice sheets and TWS from GRACE is not the primary goal of this paper. While the discussion on the mass changes already dilutes the discussion a bit, adding an extra discussion on the acceleration would weaken the focus of the paper even more.

2. Because the GIA correction that we apply is linear over the GRACE era, the uncertainty in the acceleration due to GIA is zero, while the acceleration uncertainty is likely primarily driven by decadal and multidecadal variations that will manifest as accelerations over the short data record. We do not analyze the temporal autocorrelation structure of our time series, and doing so in a proper way would be a study on its own. As a result, when we would quantify the uncertainty in the acceleration using the present approach, we'd get a very small uncertainty in the acceleration, and the resulting accelerations could easily be mis-interpreted.

Since our barystatic estimates are not to be considered as an update of previous estimates, many of which contain an in-depth analysis and tailor-made methodology to obtain individual mass changes, which is something we did not do. Therefore, we have removed the barystatic results from the abstract and the

conclusions section. As a final note, we'd like to emphasize that the underlying data is available from a repository, which means that everyone can compute the acceleration directly from the time series.

p 11 L20 : What is your indicator of "smooth temporal variations"?
This sentence is indeed not clear: what we meant was that the interannual variability in glacier and ice mass loss is small compared to the trend, and that therefore, the impact of the specific time span of the GNSS record has a limited effect on the deformation trend. We have rewritten the section to make this clear:
P13L13: The cryosphere-driven SED trends mostly show smooth spatial variations, and compared to the trend, the interannual variations are small, which implies that the specific time span of the GNSS record has a limited impact on the observed deformation rate.

p 13 L9 : Because GRACE resolution and mascon geometry is not adapted near the coast? Following that train of thought, it is unlikely that GRACE-derived TWS changes can account for the strong local variability evidenced at tide gauges.
We tried to explain a different process here that is not related to the mascon solution, but to the fact that mass changes occur on land, and the largest SED is therefore also expected to be in the middle of the region where the mass changes take place. Since tide gauges are located along the coast, they are generally not in the middle of the area where the mass change takes place (except maybe in cryosphere regions with marine-terminating glaciers), and the SED trends along the coast are thus expected to be smaller than the SED trends inland. Since we do not quantify this effect, we have removed this statement.

p 14 L10: this can easily be estimated with the application of a spatial filter on the observations (e.g. Pfeffer et al., 2017).
For some regions, especially in densely-sampled regions, such a filter would indeed provide a good estimate of long-wavelength features. However, in the regions that we refer to in this paragraph, this method comes with a problem, which is especially visible in Greenland. Here, virtually all GNSS stations are located along the coastline, which is also the region where the mass loss takes place, and the GNSS records all observe signals that are dominated by local processes, and are not representative for larger-scale processes. Applying a spatial filter here would just average the localized signals that are not representative for the whole region, and the resulting filtered mean would not be representative for the whole area. Therefore, we have decided not to implement such a filter.

p 14 L15 to 18: can you provide some quantitative information about the agreement/ disagreement between VLM observations and predictions (root mean square error, coefficient of determination, bias, maximum differences, distribution of the differences etc.)
We have added an overview of the aggregated statistics of the agreement between the model and observations to the results section.
P16L4: The mean linear trend for all 8166 stations is 0.34 mm yr$^{-1}$ with a standard deviation of 4.46 mm yr$^{-1}$, while the mean residual trend is 0.44 mm yr$^{-1}$ with a standard deviation of 4.28 mm yr$^{-1}$. The coefficient of determination (R2) of the modelled trends is just 7.5 percent. The full list of stations also contains many stations for which the MIDAS estimates very large uncertainties, sometimes exceeding 10 mm yr$^{-1}$. If we limit our station selection to those stations for which the uncertainty in the observed VLM trend is smaller than 1 mm yr$^{-1}$, which is the case for 4252 out of 8166 stations, we obtain a coefficient of determination of 34 percent and a reduction of the standard deviation from 2.30 to 1.86 mm yr$^{-1}$.

An interesting finding is that the model does not explain a large fraction of the observed trends, which changes when the trends with large standard errors are discounted. We added a remark about this in the conclusions section:
P18L28: This solid-earth deformation resulting from GIA and contemporary mass redistribution explains a substantial part of the observed GNSS trends: for all 8166 stations, we obtain a coefficient of determination of 7.5 percent. When we only consider stations for which the standard error in the observed VLM trend is smaller than 1 mm yr$^{-1}$, the coefficient of determination becomes 34 percent. This difference suggests that a nonnegligible part of the residual VLM trends can be attributed to the uncertainty in the estimated linear trends in VLM from noisy GNSS data, and that the uncertainties should not be overlooked when applying the observed and residual VLM trends to tide-gauge data.

p14 L20: why do you limit your analysis to these 15 stations? why such a restrictive selection?
Please see our response to the major point iii above on why we chose this approach.

p15 L7 to 8 "For the full model . . . GIA RSL trend" Why? How does that help with the extrapolation of non-linear processes in time?
It actually avoids the extrapolation of a non-linear process in time, as a non-linear process has been removed from the VLM observations. We have added an explanation about this:
P17L10: The 'full' model removes the spatial variations due to GIA and local VLM not related to SED, while it avoids the extrapolation of the non-linear SED signal due to contemporary mass redistribution. In the 'full' model, sea-floor deformation signals due to contemporary mass redistribution and GIA are also retained in tide-gauge records.

p 15 L10 "assuming a power law spectrum" to describe temporally correlated noise?
Yes, we have added a clarification to the text.

p 15 L11: Why do you choose the T16 subsample? How does it help to validate your approach?
Please see our response to the major point iii above on why we chose this approach.

p 16 L1-2: How are these issues resolved?
This sentence was not completely clear: we only resolve the problems due to contemporary mass redistribution. We have changed the sentence to reflect this.

p16 L9: Yes, the "sampling bias" is a huge problem that needs to be addressed. More observations are available to start with.
This sampling bias actually refers to T16 who argue that 20th-century sea-level rise is likely larger than the average at the 15 tide-gauge stations due to sea-level fingerprints and ocean dynamics. We put this sentence here, because the argumentation in T16 can be roughly summarized as follows: 1. The trend at long TG records is larger than some recent GMSL reconstructions. 2. Due to their location, we expect these tide gauges to underestimate rather than overestimate the 20th-century GMSL trend. Therefore, these observations cannot be reconciled with recent reconstructions. Here, we show that vertical land motion can explain step 1, but we do not look into step 2 here, so the discrepancy is not yet fully solved. We have changed the sentence to clarify this.

*Section 4: Conclusion*

p 16 L20: Why this disagreement? This should be discussed earlier in the manuscript.
As described above, we found a bug in our code that led to land-mass changes that were too small. After correcting this bug, the discrepancy has become smaller, but still significant. A likely candidate for this discrepancy has been found in a recent paper from Uebbing et al. (2019), who found that some of these estimates have treated the global atmospheric correction (GAC) in an inconsistent way. However, since the main focus of this paper is about local SED effects rather than updating the global sea-level budget and its contributors, we have removed this discussion from the conclusions.

p 17 L6: How do you estimated uncertainties that are not related to GIA?
The uncertainties in the other terms are all derived from the ensemble. Since the estimates of contemporary mass redistribution depend on the GIA correction applied to the GRACE data, GIA uncertainty propagates into all other quantities. Together with the GRACE measurement uncertainty and the uncertainty in the estimated VLM trends from GNSS data, this results in uncertainties for all quantities. We have added Figure 1 which gives an overview on how the ensemble is used to derive all quantities and uncertainties.

p 17 L 23: please provide metrics to illustrate the agreement/disagreement between VLM observations and predictions.

We have added these metrics to the results section. See also our response to the comments above.

p 17 L 33: The authors did not convince me that their approach avoids the bias, due to the extrapolation of non-linear VLM. It does not logically follow their assumptions and has not been evidenced in the results. However, they provide insights on the mechanisms driving recent (GRACE era) sea level changes and solid earth deformation, which is, I think, useful results.

Please see our response to the general comments above.

**Details:**

*Abstract:*
L1 to 3: rephrase the first sentence to be more readable (less "and", please)
We have updated the abstract with shorter sentences, and we have rephrased this sentence as well.

L8: "the temporal variations affect GNSS-derived VLM": temporal variations of what?
This sentence has been replaced as well.

L13 to 16: This is very confusing. Separating VLM observations in GIA and PDMR components does not separate linear from non-linear components (see major comment).
We have re-phrased this sentence as well to emphasize that we only investigate the bias due to contemporary mass redistribution.

*Introduction:*

L32 to34: see major comment
Adjusted, see above

*Data and methods:*

L10: "GIA affects GRACE observations . . .": replace by causes changes in the gravitational potential observed by GRACE satellites or equivalent to keep the same structure for each proposition of the sentence.
Fixed

L27 p3: "we bin the quantity of interest": not sure what bin means
We have re-phrased this sentence:
P5L21: We can also derive an empirical PDF from the ensemble, from which confidence intervals at any level can be computed. To compute the empirical PDF, we first sort the values from low to high. Then we define 20 bins between the 1st and 99th percentile of these values and compute the sum of the likelihood of all ensemble members that fall within each bin. The number of 20 bins is chosen as a trade-off between resolution and sample size

eq 3: why using two different variables (R and z)?
We have replaced $z$ by $R$ throughout the text.

p 7 L14: use of the term "uplift": replace by vertical displacement of the earth's surface (or equivalent)?
This sentence has been re-phrased, and we have checked that now every occurrence of the word 'uplift' refers to land coming 'up'.

Results p10 L8 and 9: "considerable" means how much?

We have rephrased the sentence and removed 'considerable':
P11L21 Figure 8 shows that, next to uplift at locations where the ice-mass loss takes place, mass changes in the cryosphere result in some far-field SED signals, including subsidence of about 0.5 mm yr$^{-1}$ around Australia, and an uplift signal with a similar magnitude in Europe and Northern Asia.

Conclusion p16 L14- 17: please reformulate to clarify
We have re-ordered the conclusions, and this sentence is not there anymore.

Figures: It is difficult to read the color scales of figure 7, 8 (bottom row), 9 and 10. Is possible to provide a bit more contrast to have a better idea of the range of variations of RSL and VLM trends? Otherwise, provide numbers in the legend or in the text.
We have updated the color scales which should now have more contrast in these figures. Alternatively, all individual data is available from the data in the repository.

**Reviewer 2**

The paper aims to develop estimates for deformation due to GIA and present-day mass redistribution with uncertainty estimates. The deformation patterns are explained and used to correct tide-gauges. It is an original idea to address a highly relevant problem. It is shown that this correction improves consistency between sea-level trends from tide
gauges and sea level reconstructions, and offers potential for better regional sea level projections. I see no problems in the methodology. Therefore I recommend the paper to be published after addressing two main comments below, and the long list of textual comments connected to these.

Correcting tide gauges only holds for the time period of the data, or the time for which the model is valid, while one of the reasons for decomposing the relative sea level rate is to make projections outside the data range. The corrections are necessarily based on limited time span and also do not include regional deformation processes which can vary in time. This is an important limitation of the paper that is not discussed well in section 3.4 and in the conclusions, see specific comments below.

A second problem in the paper is the way it is written. Logic is sometimes hard to follow, procedures are not described clearly or are not explained, the use of words that should have a precise meaning is a bit sloppy (words such as trend or linear trend, deformation, model, relative sea level or just sea level). Several examples are given below. A thorough revision of the text is necessary because it now guessing is required from the reader in several places to put the pieces of the puzzle together.

A pdf is attached with small textual comments or typos.

**Specific comments**

The abstract is a mix of describing the processes and the methodology. I suggest to move the methodology to the last paragraph where the methodology is now partly described
Thanks for this suggestion, we have re-written parts of the abstract and we have removed some of the methodological details to improve the readability.

*Introduction*

The introduction is difficult to follow because it is a mix of a background and methodology, and the description of the objective is scattered. I suggest to reorder the introduction to more clearly separate background, problem statement, methodology and application separately. Also the first part of the conclusion and the introduction should be better aligned.
We have re-written the introduction, and now we start with a statement of the problems that occurs when long tide-gauge records are corrected for VLM using short GNSS records. We have also re-written parts of the conclusions to keep the focus on the effects of SED on VLM and sea-level observations.

Suggest to add Wu and Peltier (1984) to Milne and Mitrovica (1998)
We have added the reference.

- line 3 I get the impression that sea level and relative sea level are not used consistently according to their definitions, please check
We have checked and updated the paper, and now we use global-mean sea level, geocentric sea level and relative sea level throughout the paper.

- line 15 and further: use trend or linear trend consistently

Fixed

- line 34: "to avoid this possible bias" Please make clear what bias you refer to and what part you aim to remove. If the bias refers to the (local) processes in line 21 and further then correcting for PDMR and GIA alone is not sufficient. If this bias refers to the bias due to PDMR alone (line 28 and further) it is not clear why you would remove GIA as well. Also the time-period seems relevant because you can not use the computed PDMR induced deformation beyond the period of the data.
This point was also risen by the other reviewer. We have changed the text throughout the manuscript to make clear that we only look for the bias related to contemporary mass redistribution, while other processes could also have a similar effect.

*Page 3*

- Line 6: the text contains 'estimates of GIA', 'GIA solutions','model ensemble'. Please make it more consistent.
Fixed

- Line 9: title of section 2.1 , I suggest to use something like 'prediction ensemble' because you do not actually discuss the model
Fixed: changed to 'GIA estimates'

- line 15: It seems to be partly circular reasoning when you use model ensembles scored according to fit to GNSS data to correct GNSS data. Please address this in the text.
That's true. We have added a note about this:
P5L7: Note that some circular reasoning is introduced here, as we use the same GNSS data to determine residual VLM that was used to benchmark the GIA estimates. However, since the vast majority of the benchmark data comes from paleo indicators, and the cost function used to compute the likelihood is much more sensitive to paleo rather than GNSS data, the impact on this circular reasoning on the results is limited.

*Page 4*

- Line 5: please explain why ocean bottom pressure changes are not used, because they are a form of PDMR
We have done this because the spatial variations in bottom pressure are very small compared to land-mass changes. We have added a remark:
P5L31 We only look into mass changes on land, and do not take ocean-bottom pressure changes driven by ocean dynamics into account, since its spatial variations are much smaller than the land mass changes (e.g. Watkins et al, 2015).

- Line 6: please discuss the possible effect, if any, of this filter on the final estimates, as the main interest is in deformation along the coast.
This filter reduces the leakage of mass signals between land and ocean. Therefore, the spatial distribution of mass changes in coastal locations is improved by applying this filter. Note that this filter is part of the standard processing of JPL mascons. We have added a remark about the improved spatial representation:
P7L1: For mascons that contain a coastline, a Coastline Resolution Improvement (CRI) filter has been used to prevent the leakage of gravity signals between land and ocean, which leads to a better spatial representation of mass changes in coastal regions.

*Page 5*

Potential uncertainty from determining deformation due to ice sheets, glaciers and TWS should be addressed here.
We have added a note about this uncertainty to the conclusions:

P19L26: Another limitation is that we compute SED with an elastic model that assumes a laterally uniform Earth structure. In some regions, such as West-Antarctica, elastic properties can deviate from their global-mean values and visco-elastic effects could occur on decadal time scales, which leads to additional deformation on top of the elastic response.

*Page 6*

- Line 13: It is more useful to say what is neglected: viscous effects due to PDMR changes before and during the period of interested, and where these effects could occur
We have added a remark about this, see previous comment.

- Line 15: please add a reference as there has been discussion about the methodology
We have added a reference to Milne & Mitrovica (1988), which describes the methodology that we use to compute the rotational feedback.

*Page 7*

- Line 14: how is the height anomaly defined?
With respect to the mean height of the time series. However, since we are interested in height changes, the mean height itself does not affect our results. We have added a remark:
P3L26: We express height anomalies with respect to the mean height of each time series, but since we are interested in height changes, the mean height itself does not affect our results.

Section 2.4 It would help the reader if you explain what result you are after in this section before describing possible corrections.
We have re-ordered the methods section, and we now start with an overview of the residual VLM, as that is where we are after.

*Page 9*

- line 7-15: There are probably existing studies on TWS and land mass changes that these results can be compared too.
As far as we know, almost all studies that quantity TWS changes use GRACE, and not many studies exist that quantify TWS changes from models. A study by Scanlon et al (2018) discusses the problems that models have to estimate TWS changes. We add a discussion about this in the conclusions:
P19L21: While for ice mass changes model results show good agreements with observations, model estimates of terrestrial water storage changes are still less reliable than GRACE observations (Scanlon et al., 2018), which in turn limits the ability to use models to estimate SED.

*page 11*

section 3.2, the last paragraph in the section needs a conclusion, which could perhaps be moved here from section 3.3
We have rewritten the concluding remarks of this section.

- line 18: do you mean the solid earth deformation trend from section 3.2? How do GNSS observations play a role there?
Yes, we have re-phrased this expression.

- line 20: the temporal variations are not shown in figure 11
The word temporal was indeed not correct here. We have removed it.

*page 13*

- line 3: It should be explained what kind of cryosphere changes could cause these kind of uplifts

We have re-phrased these sentences:

P13L16: These trends do not only reflect the well-known near-field uplift signals at or near the ice-mass loss locations, which dominate the VLM signal for many regions where ice mass loss occurs, but also in the far field, with notable uplift in large parts of Europe and the US, and subsidence in Australia.

*page 14*

Line 2: discussion of the uncorrected trend could be moved before the statement that the observations will be corrected.

We re-shuffled this section, and now we start with discussing the observed VLM trends before moving to the corrections.

Line 8: "even considering the uncertainty." This is ambiguous. Please make clear whether you mean that the ensemble mean is stronger or not, or whether you talk about a statistically significant increase

We have removed the statement about the uncertainty, and made clear that we discuss the ensemble mean value here.

Line 20: "partially repeat the analysis". Describe the analysis because now it is not clear what you are doing different and why in what follows.

We have removed this sentence and we expanded the description of what we have actually done.

*Page 15*

Line 1: specify what you mean by "in the vicinity" From line 5 onwards the text is very hard to follow. You need to explain that the goal is and why certain choices are made. The only explanation is that the analysis of Thompson et al. (2016) is partially repeated. Please add intepretation of the figure to line 12, now it is left to the reader. Several comments and questions on this section can be found in the pdf.

We have expanded the explanation of this section to make it better understandable.

*Page 16*

Line 1: I don't agree that both issues are resolved (same for page 17 line 32). Regional deformation such as given in page 2 line 22 will also not follow a constant trend so you can not use GNSS data or models with a shorter period than the tide gauge period and expect that extrapolation of deformation models or data works fine, or am I missing something?

We agree that this conclusion is misleading: we only considered the effects of contemporary mass redistribution. We have changed the text throughout the manuscript to make sure that it is clear that any other non-linear process is still in the data.

Line 10: "the gap discussed by T16" specify which gap for readers that have not read that paper

We have added a clarification:

P18L14: Note that T16 also argues that averaging the linear trend in sea level from these long tide gauges likely underestimates the global-mean sea-level trend due to the spatial patterns associated with ice-mass loss and ocean dynamics. Here, we do not consider this spatial sampling bias, so the gap between the long tide-gauge records and global-mean sea-level reconstructions discussed by T16 cannot yet be fully reconciled from these results.

*Page 17*

Line 25: "uncertain GIA contribution" explain if this means that the uncertainty in the GIA models ensemble is underestimated

Yes, that is indeed what we wanted to stated. We have added a remark:

P19L3: A likely candidate for this residual trend is the uncertain GIA contribution: the global model that we use does not account for lateral variations in the mantle viscosity structure and is not optimized for a specific region,

and uncertainties in the deglaciation history that are not fully represented in the GIA ensemble could lead to an underestimation of the uncertainty in formerly glaciated regions.

Please also note the supplement to this comment
We have addressed all typos and unclear statements that were brought forward in the supplement.

**References**

Dangendorf, S., Marcos, M., Wöppelmann, G., Conrad, C. P., Frederikse, T., & Riva, R. (2017). Reassessment of 20th century global mean sea level rise. Proceedings of the National Academy of Sciences, 114(23), 5946–5951. https://doi.org/10.1073/pnas.1616007114

Hay, C. C., Lau, H. C. P., Gomez, N., Austermann, J., Powell, E., Mitrovica, J. X., … Wiens, D. A. (2017). Sea Level Fingerprints in a Region of Complex Earth Structure: The Case of WAIS. Journal of Climate, 30(6), 1881–1892. https://doi.org/10.1175/JCLI-D-16-0388.1

Milne, G. A., & Mitrovica, J. X. (1998). Postglacial sea-level change on a rotating Earth. Geophysical Journal International, 133(1), 1–19. https://doi.org/10.1046/j.1365-246X.1998.1331455.x

Scanlon, B. R., Zhang, Z., Save, H., Sun, A. Y., Müller Schmied, H., van Beek, L. P. H., … Bierkens, M. F. P. (2018). Global models underestimate large decadal declining and rising water storage trends relative to GRACE satellite data. Proceedings of the National Academy of Sciences, 115(6), E1080–E1089. https://doi.org/10.1073/pnas.1704665115

Schaeffer, N. (2013). Efficient spherical harmonic transforms aimed at pseudospectral numerical simulations. Geochemistry, Geophysics, Geosystems, 14(3), 751–758. https://doi.org/10.1002/ggge.20071

Thompson, P. R., Hamlington, B. D., Landerer, F. W., & Adhikari, S. (2016). Are long tide gauge records in the wrong place to measure global mean sea level rise? Geophysical Research Letters, 43(19), 10,403-10,411. https://doi.org/10.1002/2016GL070552

Uebbing, B., Kusche, J., Rietbroek, R., & Landerer, F. W. (2019). Processing choices affect ocean mass estimates from GRACE. Journal of Geophysical Research: Oceans. https://doi.org/10.1029/2018JC014341

Wöppelmann, G., Marcos, M., Santamaría-Gómez, A., Martín-Míguez, B., Bouin, M.-N., & Gravelle, M. (2014). Evidence for a differential sea level rise between hemispheres over the twentieth century. Geophysical Research Letters, 41(5), 1639–1643. https://doi.org/10.1002/2013GL059039

Watkins, M. M., Wiese, D. N., Yuan, D.-N., Boening, C., & Landerer, F. W. (2015). Improved methods for observing Earth's time variable mass distribution with GRACE using spherical cap mascons. Journal of Geophysical Research: Solid Earth, 120(4), 2648–2671. https://doi.org/10.1002/2014JB011547

Wu, P., & Peltier, W. R. (1984). Pleistocene deglaciation and the Earth's rotation: A new analysis. Geophysical Journal International, 76(3), 753–791. https://doi.org/10.1111/j.1365-246X.1984.tb01920.x

Zemp, M., Huss, M., Thibert, E., Eckert, N., McNabb, R., Huber, J., … Cogley, J. G. (2019). Global glacier mass changes and their contributions to sea-level rise from 1961 to 2016. Nature, 568(7752), 382–386. https://doi.org/10.1038/s41586-019-1071-0